# A modular self-adjuvanting cancer vaccine combined with an oncolytic vaccine induces potent antitumor immunity

Krishna Das[1,2,8], Elodie Belnoue[3,4,8], Matteo Rossi[3,4], Tamara Hofer[1,2], Sarah Danklmaier[1,2], Tobias Nolden[4,5], Liesa-Marie Schreiber[1,2], Katharina Angerer[1,2], Janine Kimpel [2], Sandra Hoegler[6], Bart Spiesschaert[4,5], Lukas Kenner [6,7], Dorothee von Laer [2], Knut Elbers[4,5], Madiha Derouazi [3,4✉] & Guido Wollmann [1,2✉]

Functional tumor-specific cytotoxic T cells elicited by therapeutic cancer vaccination in combination with oncolytic viruses offer opportunities to address resistance to checkpoint blockade therapy. Two cancer vaccines, the self-adjuvanting protein vaccine KISIMA, and the recombinant oncolytic vesicular stomatitis virus pseudotyped with LCMV-GP expressing tumor-associated antigens, termed VSV-GP-TAA, both show promise as a single agent. Here we find that, when given in a heterologous prime-boost regimen with an optimized schedule and route of administration, combining KISIMA and VSV-GP-TAA vaccinations induces better cancer immunity than individually. Using several mouse tumor models with varying degrees of susceptibility for viral replication, we find that priming with KISIMA-TAA followed by VSV-GP-TAA boost causes profound changes in the tumor microenvironment, and induces a large pool of poly-functional and persistent antigen-specific cytotoxic T cells in the periphery. Combining this heterologous vaccination with checkpoint blockade further improves therapeutic efficacy with long-term survival in the spectrum. Overall, heterologous vaccination with KISIMA and VSV-GP-TAA could sensitize non-inflamed tumors to checkpoint blockade therapy.

---

[1] Christian Doppler Laboratory for Viral Immunotherapy of Cancer, Medical University of Innsbruck, Innsbruck, Austria. [2] Institute of Virology, Medical University of Innsbruck, Innsbruck, Austria. [3] AMAL Therapeutics, Geneva, Switzerland. [4] Boehringer Ingelheim International GmbH, Ingelheim, Germany. [5] ViraTherapeutics GmbH, Innsbruck, Austria. [6] Unit of Laboratory Animal Pathology, Institute of Pathology, University of Veterinary Medicine Vienna, Vienna, Austria. [7] Department of Experimental Pathology, Medical University of Vienna, Vienna, Austria. [8] These authors contributed equally: Krishna Das, Elodie Belnoue. ✉email: madiha.derouazi@boehringer-ingelheim.com; guido.wollmann@i-med.ac.at

In recent years, the long pursued concept of immunotherapy rose from promise to effective treatment modality for a selection of tumors[1,2]—largely driven by the success of immune checkpoint inhibitors (CPI) and adoptive cell therapies[3]. However, not all types of tumors and/or patients respond to these therapies, due to inefficient and low immune cell infiltration in the tumors, tumor heterogeneity, or multiple immune escape mechanisms. Those limitations can be addressed by therapeutic cancer vaccines on one side and oncolytic virotherapy on the other, highlighting the major potential of combining such modalities.

The KISIMA vaccine platform, a chimeric recombinant protein[4], is composed of three key elements within a single protein: first, a rationally designed Multi-Antigenic Domain (Mad) containing multiple relevant tumor antigens with different cytotoxic T lymphocytes (CTL) and helper T cell stimulating epitopes for various human leukocyte antigen (HLA) restrictions. Second, a Cell Penetrating Peptide (CPP) that ensures antigen delivery and promotes an integrated CTL and helper T cell response[5,6]. Third, a peptide agonist (TLRag) for Toll like receptor (TLR)−2 and TLR-4, which confers self-adjuvanting properties to the vaccine. The vaccines developed using this platform deliver multiple epitopes from various antigens to address potential antigen loss in tumors. Additionally, cytotoxic T cells induced by the KISIMA vaccine have a high frequency of memory precursors[4], which generate a long-lasting memory pool[7].

Oncolytic viruses (OV) exert their therapeutic effects by a number of interlinked mechanisms. However, susceptibilities for prolonged oncolytic action vary between tumors[8] and the associated intratumoral inflammation as well as general immune activation constitute a critical additional mode of action of oncolytic virotherapy[9]. In addition, the immunogenic nature of oncolysis can facilitate the release and recognition of tumor-associated antigens[10]. Last but not least, oncolytic viruses are able to amplify their therapeutic signals once delivered to a permissive tumor[11]. As with vaccines, most OV monotherapies show clinical activities that fall behind their preclinical promise[12]. One potential limitation is inherently linked to the strong immune activation that comes with two dominant antiviral forces, an initial innate and a subsequent adaptive response[13,14], although these very same mechanisms may also counter tumor-associated immune suppression[15,16]. Arming OVs with antigens associated with the tumor can additionally enhance the tumor-specific T cell portion and therefore positively affect the balance of antitumor versus antiviral immune responses[17].

VSV-GP is a chimeric vesicular stomatitis virus (VSV) variant, pseudotyped with the LCMV-GP protein[18], resulting in abrogated neurotoxicity while maintaining the lytic potency and broad tumor tropism of the parental virus[19–21]. As an oncolytic agent, VSV-GP induces strong innate and adaptive immune responses in permissive tumors[16]. Conversely, as a vaccine vector armed with specific target antigens, VSV-GP elicits a strong and lasting CTL and antibody response in a homologous prime-boost setting[18].

A heterologous prime-boost regimen combining a non-viral cancer vaccine with an oncolytic vaccine platform may hold the key for both modalities to overcome their monotherapeutic limitations[22]. The alternating application of two vaccine platforms expressing shared tumor antigens results in a shift from a dominantly antiviral CTL response to an enhanced antitumor response[23], while in permissive tumors the lytic component with an associated release of danger- or pathogen-associated molecular patterns (DAMPs or PAMPs, respectively) favorably affects the tumor microenvironment (TME)[24]. While the concept has been presented in several preclinical studies[25,26], and clinical testing has commenced in recent years[17], the mechanistic dissection of such combinations has been limited.

Here we show that such heterologous oncolytic vaccination not only leads to a pro-therapeutic TME repolarization and enhances quantity of antigen-specific T cell responses, but also that the combination of KISIMA-TAA and VSV-GP-TAA affects the quality of the tumor-specific T cell pool on several levels. Using various murine tumor models representing different antigen classes (model antigen Ovalbumin or OVA, neoantigens Adpgk (ADP-dependent glucokinase) and Reps1 (RalBP1-associated Eps domain-containing protein 1), and oncoviral antigen HPV-E7), we dissect the induced immune components in several compartments addressing immunogenicity and efficacy.

## Results

**Heterologous prime-boost with KISIMA-TAA and VSV-GP-TAA induces long-term immunity.** To characterize the heterologous combination of KISIMA-TAA with VSV-GP-TAA oncolytic vaccine, we first investigated the schedule of administration using OVA. Priming with subcutaneous injection of KISIMA-OVA followed by intramuscular boost with VSV-GP-OVA elicited higher proportion (frequency and number) of circulating OVA-specific CD8+ T cells compared to VSV-GP-OVA prime and KISIMA-OVA boost (Fig. 1a). Importantly, the OVA-specific T cell population further expanded after a second boost with KISIMA-OVA but not with VSV-GP-OVA (Fig. 1a). Thus, for subsequent experiments an alternating regimen was selected starting with KISIMA-TAA priming, VSV-GP-TAA boost followed with a second KISIMA-TAA boost (KVK regimen). Next, different routes of virus administration were compared. A single intravenous (i.v.) administration of VSV-GP-OVA induced the highest frequency of antigen-specific CD8+ T cell response compared to intraperitoneal (i.p.), subcutaneous (s.c.) or intramuscular (i.m.) injection for both OVA (Supplementary Fig. 1a) and VSV-N (Supplementary Fig. 1b) antigens. Subsequently, immune response elicited upon boost with either i.v. or i.m. VSV-GP-OVA was assessed using the KVK regimen. Intravenous boost resulted in a significantly higher proportion of OVA-specific peripheral CD8+ T cells, which further expanded following the KISIMA-OVA boost. 140 days post prime, OVA-specific T cells were still present in the i.v. group suggesting the formation of immunological memory (Fig. 1b). Consistently, a higher number of OVA-specific CD8+ T cells were found in spleen (Fig. 1c) and bone marrow (Fig. 1d) following KVK vaccination using i.v. in contrast to i.m. route. Since the phenotypical composition in general and the antigen-specific memory precursor effector cells expressing CD127 in particular are important for generation of long-lasting memory[27], the presence of CD127+ memory precursors and KLRG1+ effector cells among the antigen-specific T cells was assessed (Supplementary Fig. 1c). Systemic administration of VSV-GP-OVA in KVK regimen resulted in a higher number of both memory and effector OVA-specific CD8+ T cells in circulation (Fig. 1e), spleen (Fig. 1f) and bone marrow (Fig. 1g) compared to i.m. immunization, confirming its ability to generate immunological memory.

The immunogenicity of KVK regimen was further assessed against neoantigens and viral oncoprotein as target antigens. For targeting neo-epitopes, KISIMA-Mad24, a KISIMA-derived vaccine bearing the previously described neoantigens Adpgk (ADP-dependent glucokinase) and Reps1 (RalBP1-associated Eps domain-containing protein 1) which are expressed in the murine colorectal carcinoma model MC-38[28] and the corresponding VSV-GP-Mad24 were used. Additionally, KISIMA-HPV bearing HPV-derived E7 oncoprotein as antigen was used in combination with VSV-GP-HPV. The latter contained HPV-derived E7 and in addition E6 and E2. Priming with KISIMA followed with an i.v. VSV-GP-TAA boost elicited the highest frequency of antigen-specific T cells in the periphery for both

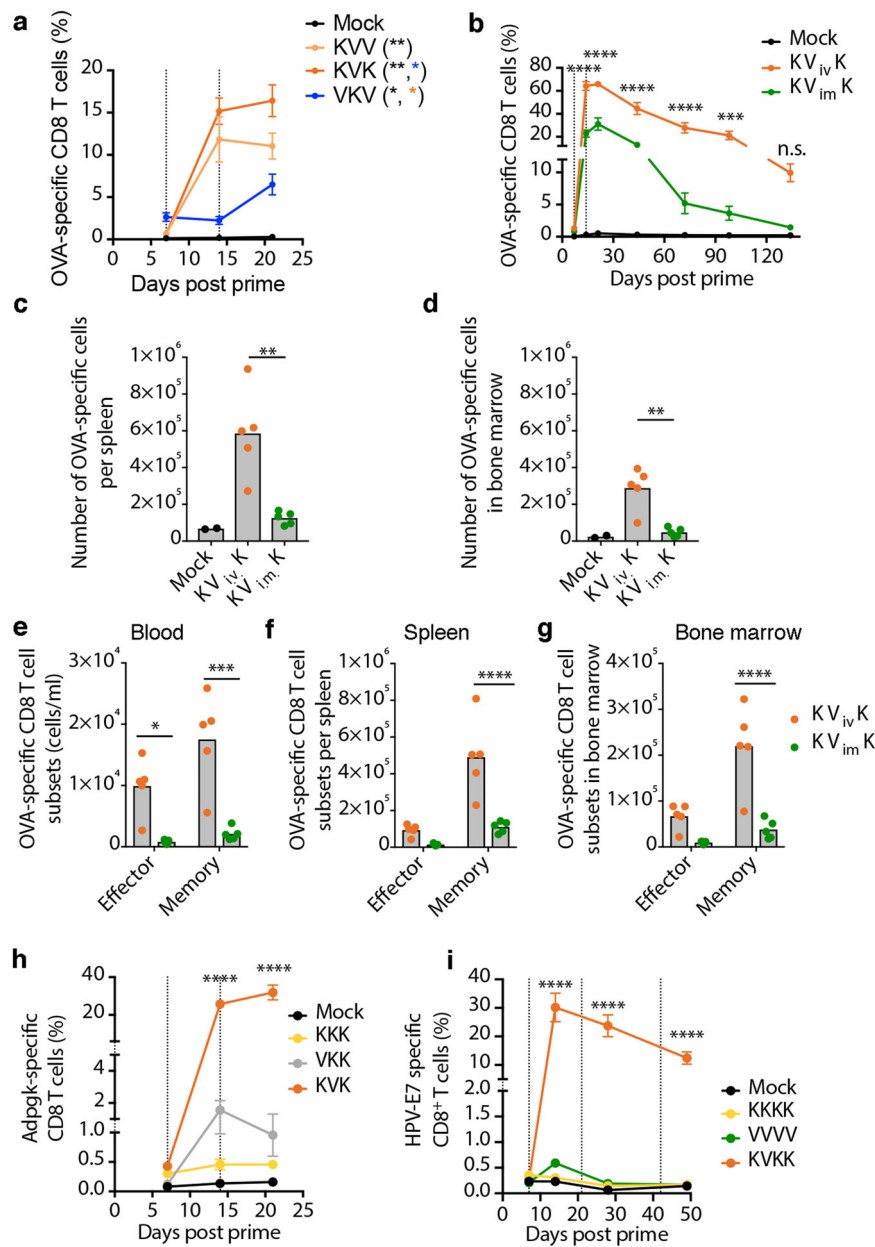

**Fig. 1 Heterologous prime-boost vaccination with KISIMA-TAA and VSV-GP-TAA is superior to homologous vaccination with either vaccine platform.**
**a** C57BL/6J mice ($n = 5$) were immunized against ovalbumin with either KISIMA-OVA given s.c. (K) or VSV-GP-OVA (V) administered i.m. on days 0, 7, and 14 (dotted lines) or left untreated (mock). The fraction of OVA-specific cells among CD8+ T cells in peripheral blood is shown. Two-way ANOVA with Tukey's multiple comparisons (*, $p < 0.05$; **, $p < 0.01$). **b-g** C57BL/6J mice were immunized with KISIMA-OVA (s.c.) on day 0 and 14 and VSV-GP-OVA was administered either i.m. or i.v. on day 7 as indicated by dotted lines ($n = 5$ per treatment group, $n = 2$ for mock). **b** The frequency of OVA-specific CD8+ T cells in circulation is depicted. Two-way ANOVA with Sidak's multiple comparison (**, $p < 0.01$). **c, d** The number of OVA-specific CD8+ T cells in the (**c**) spleen and (**d**) bone marrow of immunized mice was measured 19 weeks post prime. One-way ANOVA with Tukey's multiple comparison (**, $p < 0.01$). **e-g** The number of OVA-specific CD8+ T cells with effector (KLRG1+CD127−) and memory precursor (CD127+) phenotype in (**e**) peripheral blood (cells/ml), (**f**) spleen and (**g**) bone marrow 19 weeks after prime is shown. Two-way ANOVA with Sidak's multiple comparison (****, $p < 0.0001$). (**h, i**) C57BL/6J mice were vaccinated against (**h**) Adpgk and Reps1 neoantigens or (**i**) HPV-E7 with either (**h**) KISIMA-Mad24 or (**i**) KISIMA-HPV given s.c. (K) or (**h**) VSV-GP-Mad24 or (**i**) VSV-GP-HPV administered i.v. (V) on day 0 and as indicated with dotted lines. The frequency of circulating CD8+ T cells specific for (**h**) Adpgk ($n = 5$ per group) and (**i**) HPV-E7 ($n = 7$ per group) is depicted. Two-way ANOVA (**h**) and one-way ANOVA (**i**) with Tukey's multiple comparison was performed (*** $p < 0.001$; ****$p < 0.0001$) and significance compared to mock is shown. All data shown as mean ± SEM. Studies (**a, c-g**) were performed once, studies (**b, i**) were independently repeated once, study (**h**) was repeated once with the listed groups except VKK, which was only included once. Source data and p-values are provided in the Source Data File.

antigen models (Fig. 1h, i). Targeting E7, heterologous KVK vaccination resulted in significantly higher antigen-specific T cell responses compared to both homologous vaccinations, inducing an over 30-fold increase in circulating HPV-E7-specific CD8+ T cell

frequency (Fig. 1i). Though subsequent boosting with KISIMA-HPV did not further increase HPV-E7-specific CD8+ cells frequency, it did prevent them from undergoing dramatic contraction. Consistent with the OVA model, HPV-specific CD8+ T cells induced by

heterologous prime-boost vaccination as shown in Supplementary Fig. 2a persisted in the periphery (Supplementary Fig. 2b), bone marrow (Supplementary Fig. 2c) and spleen (Supplementary Fig. 2d) for up to 5 weeks after last immunization and displayed an effector memory phenotype (Supplementary Fig. 2e–g).

Comparing antiviral with anti-target immune responses, KVV heterologous prime-boost treatment (Supplementary Fig. 2h) not only enhanced tumor antigen-specific immunity but also dampened the antiviral response (Supplementary Fig. 2i) compared to homologous VSV-GP-OVA vaccination (VVV) or priming with VSV-GP-OVA (VKV). In addition, an inverse correlation between proportion of virus- and OVA-specific CTLs in blood was observed (Supplementary fig. 2j). This reversal in the ratio of antitumor and antiviral T cells was also reproduced for the tumor antigen E7 (Supplementary Fig. 2k). Thus, heterologous prime-boost vaccination using KVK regimen induces a potent CD8+ T cell response against model, tumor-associated and tumor-specific antigens and favors the development of immunological memory while dampening antiviral immunity.

**Priming with KISIMA-TAA improves functionality of tumor-specific T cells.** Some key properties of cancers are immune exclusion and suppression, which allow tumor cells to counter cytotoxic CD8+ T cell infiltration and function. Thus, we assessed the ability of therapeutic KVK heterologous prime-boost vaccination to overcome this constraint in the immunologically 'cold' TC-1 tumor model. TC-1 cells are transformed murine lung epithelial cells expressing the HPV-derived oncoproteins E6 and E7[29]. Once the tumors were palpable, mice were primed with KISIMA-HPV or VSV-GP-HPV followed 7 days later with a VSV-GP-HPV boost. Tumor-infiltrating cells were analyzed one week after boost (Fig. 2a). Consistent with the results in non-tumor-bearing animals, KISIMA-HPV prime followed by VSV-GP-HPV boost resulted in significantly higher frequency (Fig. 2b) and absolute numbers (Fig. 2c) of HPV-E7-specific CD8+ T cells in the periphery, compared to homologous VSV-GP-HPV treatment. Both vaccination regimens were able to induce high infiltration of CD8+ T cells within the tumor, about 60% of which were found to be HPV-E7-specific by multimer staining (Fig. 2d, e). In contrast to the periphery, there were no differences within the tumor between the two vaccine regimens in HPV-E7-specific CD8+ T cells frequency (Fig. 2d) and numbers (Fig. 2e). As the immunosuppressive tumor microenvironment is well known to induce a rapid exhaustion of T cells, we next assessed the phenotype of circulating and tumor-infiltrating antigen-specific CD8+ T cells. While only a small portion of HPV-E7-specific CTLs displayed an exhausted phenotype in the periphery (Fig. 2f), characterized by the expression of PD-1 and Tim-3, the majority of tumor-infiltrating CD8+ T cells expressed both markers - suggesting their exhaustion (Fig. 2g). Interestingly, a higher proportion of intratumoral PD-1+Tim-3+ CD8+ T cells in KV vaccinated mice still expressed the early activation marker KLRG1, suggesting a less advanced exhaustion status compared to the homologous VV treated mice. Since T cell exhaustion is a progressive process, which initiates with the expression of markers and continues with loss of function and eventually cell death, we assessed CD8+ T cells functionality by measuring cytokine secretion after ex vivo restimulation. A significantly higher proportion of splenic HPV-E7-specific CD8+ T cells in KV vaccinated mice expressed IFN-γ, TNF-α and the degranulation factor CD107a compared to VV treated mice (Fig. 2h). Additionally, higher frequency of granzyme B producing CTLs were detected in KV vaccinated mice compared to VV vaccinated mice (Fig. 2j). In accordance to the results from the spleen, KV vaccination induced a significantly higher proportion of multifunctional

HPV-E7-specific CD8+ T cells in the tumor compared to VV treatment, in particular IFN-γ+TNF-α+CD107a+ triple-positive cells (Fig. 2i), highlighting a highly cytotoxic, less exhausted phenotype of KV elicited antigen-specific CD8+ T cells.

In addition, the virus-specific immune response was also monitored in the periphery and within the tumor (Supplementary Fig. 3a–e). Similarly to the HPV-specific CD8+ T cells response, a small proportion of peripheral antiviral CD8+ T cells expressed exhaustion markers or secreted cytokines (Supplementary Fig. 3f, h). The differences in peripheral responses (Supplementary Fig. 3b, c) did not correlate with the intratumoral response (Supplementary Fig. 3d, e), where higher numbers of VSV-N-specific CD8+ T cells were observed in VV compared to KV vaccinated mice (Supplementary Fig. 3d, e). However, intratumoral VSV-N-specific CD8+ T cells mostly expressed PD-1 on their cell surface (Supplementary Fig. 3g) and showed low functionality (Supplementary Fig. 3i), suggesting that they were bystander cells.

Overall, priming with KISIMA-HPV and boosting with VSV-GP-HPV not only supports induction of higher magnitude of tumor-specific CD8+ T cells, but also promotes their recruitment into the tumor and enhances their functionality compared to homologous viral vaccination.

**Heterologous vaccination reverses immunosuppression in TME.** After heterologous KV vaccination, dramatic changes in the TC-1 TME were observed upon transcriptome analysis (Supplementary Data 1 and Fig. 3a, b). 64.9% of all panel genes were upregulated in KV treated tumors, compared to 36.8% after homologous VV vaccination; indicating stronger activation of multiple immune pathways (Supplementary Data 2 and Fig. 3a). While 244 of these genes could be attributed to the immune activating effects of VSV-GP-HPV, a set of 243 genes was upregulated only in the heterologous vaccination group (Supplementary data 2 and Fig. 3c). The genes uniquely upregulated in KV treatment are involved in both innate and adaptive immune responses (Supplementary Table 1). Interestingly, heterologous vaccination also negatively regulated the expression of 35 genes (Supplementary data 2 and Fig. 3b, d) including *Cdkn1a* and *Msln* which are involved in cancer progression (Supplementary Table 2). In addition, heterologous KV vaccination activated multiple immune genes associated with cytotoxic T cells (Fig. 3e), dendritic cells (DCs) (Fig. 3g), cytokines (Fig. 3f), chemokines (Fig. 3h) and antigen processing and presentation (Fig. 3i). Hierarchical clustering revealed that tumors from mice receiving a specific vaccine combination had a similar transcriptome and thus were more likely to cluster together. Biologically, increased CTL infiltration along with elevated levels of cytotoxic genes such as granzymes (*Grzma*, *Grzmb* and *Grzmk*) and perforin (*Prf1*) (Fig. 3e) and antigen presentation (Fig. 3i) suggested enhanced tumor cell killing as a result of heterologous vaccination. Besides, more genes indicative of DC function and maturation including cross-presentation were upregulated in heterologous treated tumors (Fig. 3g). All vaccine combinations upregulated components of the antigen processing machinery but homologous VV vaccination had a stronger impact on genes encoding MHC I and MHC II molecules; whereas KV vaccination positively regulated non-classical MHCs (Fig. 3i).

Both proinflammatory and anti-inflammatory cytokines were upregulated as a result of the immune activating effect of KV vaccination, including elevated levels of type I and type II interferons (Fig. 3f). Also, cytokines such as *Ifng* and *Tnf* important for T cell effector functions were elevated in TC-1 tumors after heterologous vaccination (Fig. 3f). Importantly, some of the cytokines and chemokines upregulated in the tumor were also elevated in the plasma of mice receiving heterologous

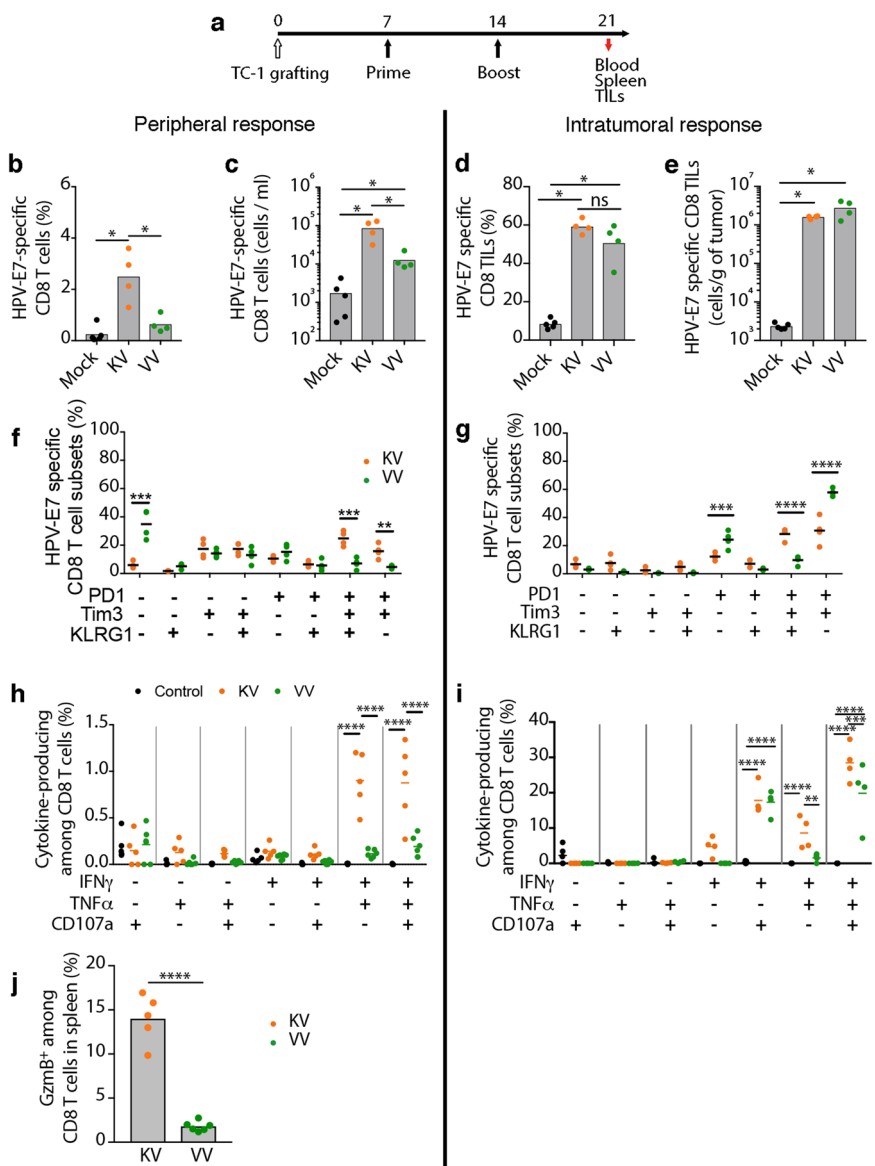

**Fig. 2 Priming with KISIMA-HPV improves functionality of intratumoral HPV-specific T cells. a–j** C57BL/6J mice were injected s.c. with TC-1 cells on day 0 and vaccinated with KISIMA-HPV (s.c.) or VSV-GP-HPV (i.v.) on day 7 and 14. Blood, spleen and tumors were harvested on day 21 for flow cytometric analysis. **a** Schematic of experimental plan. **b, d** Frequency and (**c, e**) number of HPV-E7-specific CD8+ T cells were measured by flow cytometry in (**b, c**) blood and in (**d, e**) tumors. Two-tailed Mann–Whitney test (*, $p < 0.05$). **f, g** Proportions of (**f**) peripheral and (**g**) intratumoral HPV-E7-specific CD8+ T cells expressing activation and exhaustion markers are depicted ($n = 5$ for mock, $n = 4$ for each treatment group). Two-way ANOVA with Sidak's multiple comparison (***, $p < 0.001$; ****, $p < 0.0001$). **h, i** Frequencies of HPV-E7-specific CD8+ T cells secreting different cytokines upon ex vivo restimulation among (**h**) splenic ($n = 5$ for mock and KV, $n = 6$ for VV) and (**i**) intratumoral ($n = 5$ for mock, $n = 4$ for each treatment group) CD8+ T cells are shown. Two-way ANOVA with Sidak's multiple comparison (**, $p < 0.01$; ***, $p < 0.001$; ****, $p < 0.0001$). **j** Frequencies of granzyme B expressing cells among splenic CD8+ T cells is shown ($n = 5$ for KV and $n = 6$ for VV). One-way ANOVA with Tukey's multiple comparisons (****$p < 0.0001$). All data shown as mean ± SEM. Studies (**b–e, j**) were repeated once. Studies (**f-i**) were performed once. Source data and p-values are provided in the Source Data File.

KV vaccine, including increased levels of IFN-γ, CCL5, CXCL10, CCL2, IL-6, CXCL1, and IL-1β one day after VSV-GP-HPV boost (Fig. 4a).

Observations from transcriptome analysis were further supported by the analysis of the number (Fig. 4b) and types (Fig. 4c, Supplementary Fig. 4) of tumor-infiltrating leukocytes (TILs). TILs from untreated TC-1 tumors are predominantly (>80%) composed of immunosuppressive cells such as M2-like tumor-associated macrophages (TAM-2) and myeloid-derived suppressor cells (MDSCs), while T cells constitute only 1% of the infiltrate (Fig. 4c). Therapeutic vaccination induced deep changes

in the TILs, with a striking influx of both CD8+ and CD4+ T cells populations and a drastic decrease of TAM-2, resulting in an enhanced TAM-1/TAM-2 ratio suggesting repolarization. Furthermore, heterologous KV vaccination promoted the strongest influx of CD8+ T cells (>25%) (Fig. 4c). Thus, while both vaccination regimens promoted trafficking of immune cells into the tumor, KV vaccination attracted the highest proportion of CTLs, CD4+ T helper cells and increased TAM-1/TAM-2 ratio thereby favorably remodeling the TME.

Next, immunohistochemistry was performed to confirm the location of immune infiltrates. CD8 staining of tumors harvested

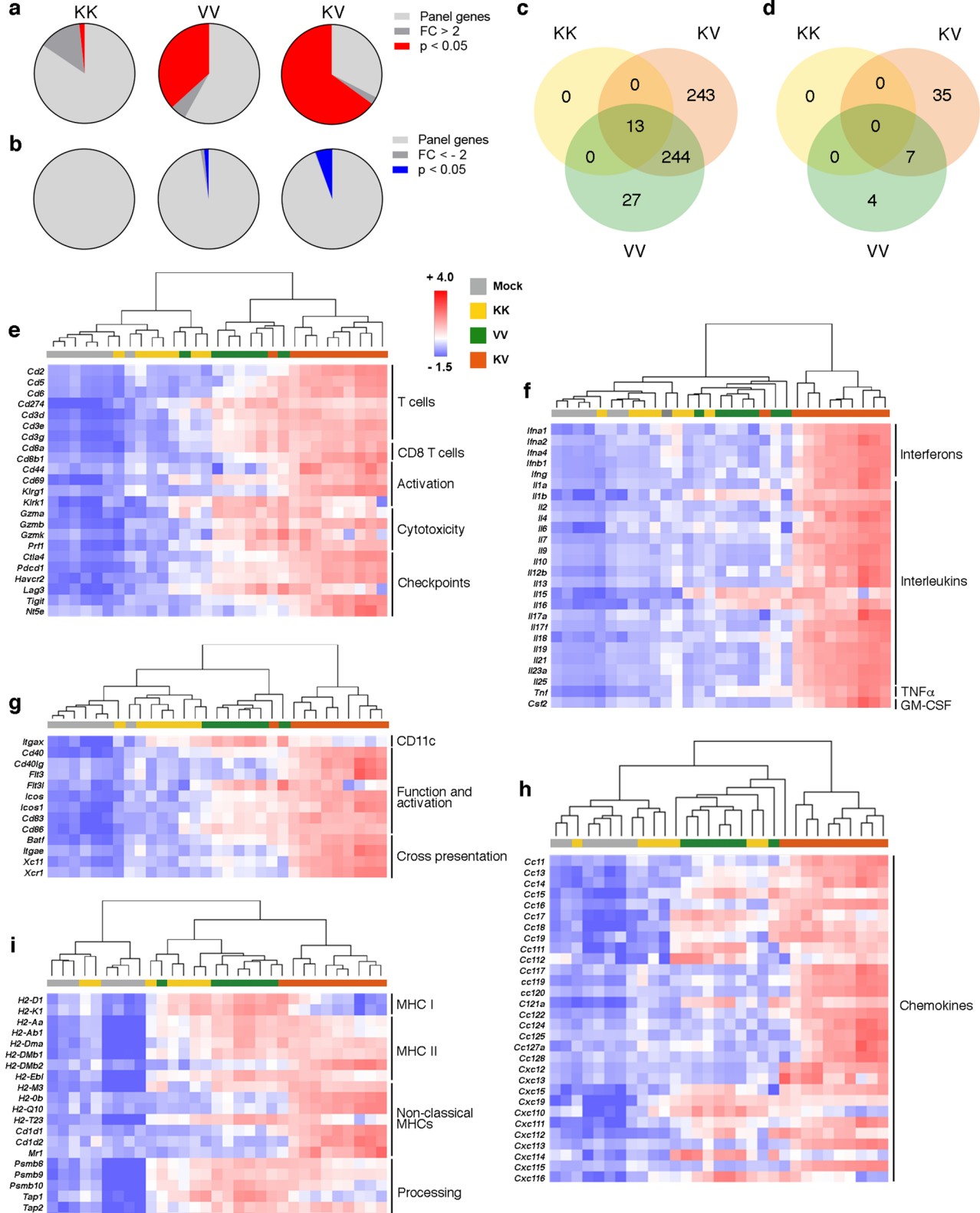

9 days post boost confirmed the general immune-excluded phenotype of untreated TC-1 tumors with few CD8+ T cells confined to the tumor margin (Fig. 4d). While CD8+ T cell infiltration increased with homologous KK and VV vaccine regimen, the heterologous combination KV displayed a massive cytotoxic T cell presence in the deepest parts of the tumor.

**Efficacy and synergy with checkpoint blockade**. Susceptibility to oncolytic viruses varies between tumors, and continued viral propagation is often limited. Heterologous prime-boost with oncolytic vaccines may address such limitations in tumors with known antigenic targets. Hence, heterologous KISIMA-TAA/ VSV-GP-TAA prime-boost was assessed in a selection of tumor

**Fig. 3 Gene signatures after heterologous vaccination indicate strong immune activation in treated tumors. a–i** C57BL/6 J mice bearing TC-1 tumors were immunized as in Fig. 2 or left untreated (mock). Tumors were harvested on day 23 post tumor implantation for transcriptome analysis using NanoString® technology ($n = 7$ for mock, KK, and VV, $n = 10$ for KV). **a, b** Gene expression in TC-1 tumors from each vaccination group was normalized to mock tumors and the proportion of (**a**) significantly upregulated (fold change [FC] >2 and $p < 0.05$) and (**b**) significantly downregulated (negative reciprocal of FC < −2 and $p$-value < 0.05) genes is displayed. **c, d** Venn diagrams depict the total number of significantly (**c**) upregulated and (**d**) downregulated genes after different vaccine regimens and the overlap between each gene set. **e–i** Heatmaps display relative gene expression as z-scores (scaled to each gene) and hierarchichal clustering (Euclidean distance, average linkage) was applied to sample data with each column representing one individual tumor. Expression of typical genes associated with (**e**) cytotoxic T cells, (**f**) cytokines, (**g**) dendritic cells, (**h**) chemokines and (**i**) antigen presentation is shown. 7–10 mice analyzed for each treatment group, $p$-values were calculated using two-tailed t test and false discovery rate (FDR) adjusted $p$-values calculated using Benjamini–Yekutieli procedure are reported. The study as shown was performed once. Groups mock and KV were repeated in a separate study. Source data are provided in the Source Data File.

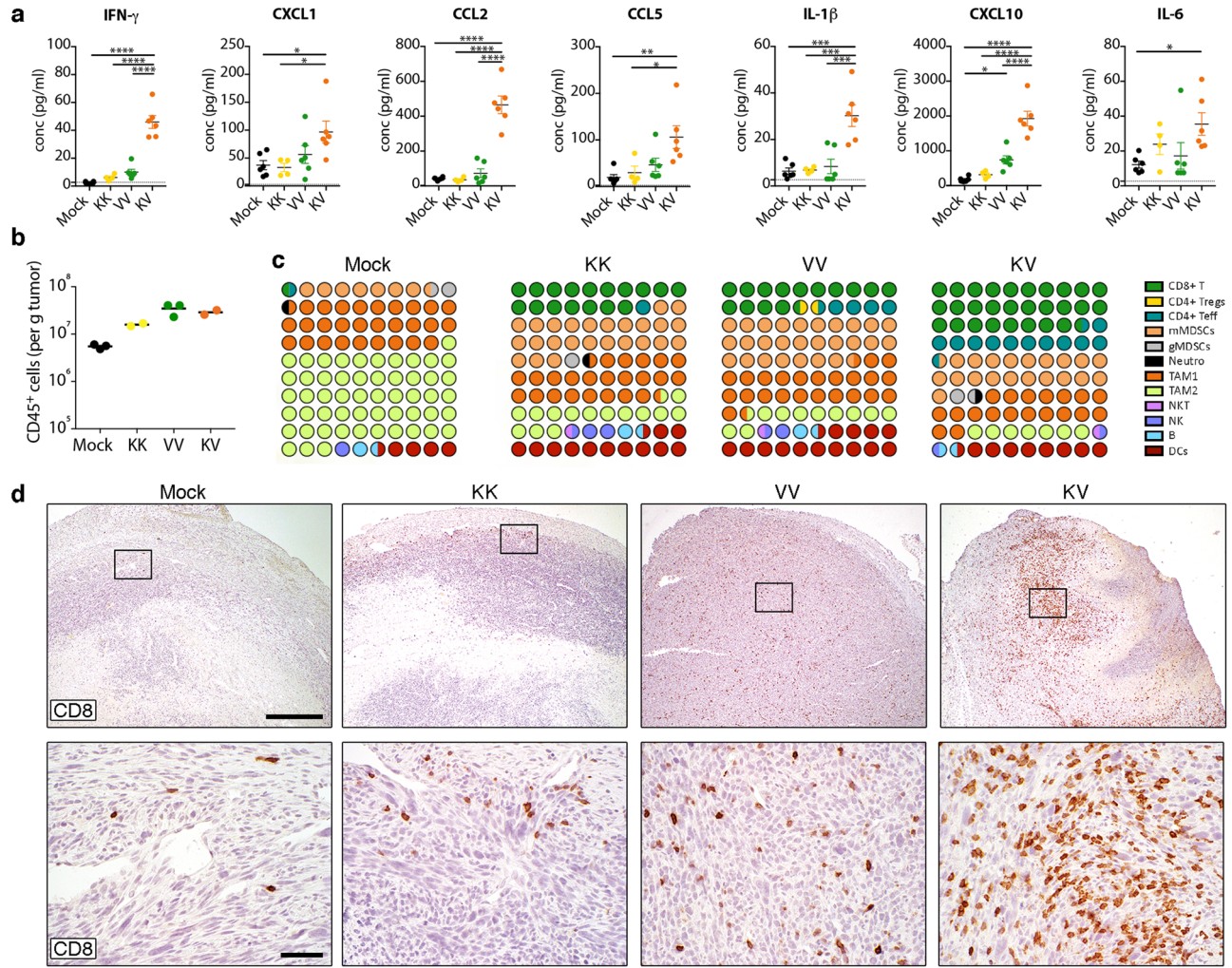

**Fig. 4 Remodeling of immunosuppressive tumor microenvironment (TME) after heterologous prime-boost vaccination.** Mice bearing palpable TC-1 tumors were immunized as in Fig. 2. **a** Cytokine and chemokine levels in plasma were quantified on day 15 and are shown as mean ± SEM ($n = 6$ for mock, VV, and KV, $n = 4$ for KK). One-way ANOVA with Tukey's multiple comparison (*$p < 0.05$, ** $p < 0.01$; ***$p < 0.001$, ****$p < 0.0001$). Dotted lines indicate the limit of quantification. **b, c** Tumors were harvested on day 26 and tumor-infiltrating leukocytes were characterized by flow cytometry ($n = 3$ for mock and VV, $n = 2$ for KK and KV). **b** Total CD45+ leukocytes (mean) and (**c**) relative proportions of various immune cell subsets among all leukocytes is shown. **d** Representative immunohistochemistry images show T cell infiltration (CD8) in TC-1 tumors day 23 post tumor implantation after different vaccinations. Lower panel shows higher magnification view of boxed area from upper panel. Scale bars: 500 µm upper row, 50 µm lower row. Study (**a**) was performed twice, studies (**c, d**) were performed once. Source data and $p$-values are provided in the Source Data File.

models that show resistance or very limited responses to oncolytic VSV-GP monotherapy. Murine lymphoma cells E.G7-OVA are resistant to VSV-GP induced oncolysis in vitro (Supplementary Fig. 5a). TC-1 and B16-OVA tumors are susceptible to infection and lysis in vitro, but are protected by IFN-mediated innate antiviral responses (Supplementary Fig. 5b, c). MC-38 tumors on the other hand show full in vitro susceptibility even in the presence of IFN, indicating impaired antiviral defense (Supplementary Fig. 5d). Of note, this in vitro susceptibility does not translate to efficacy of single VSV-GP application in vivo (Supplementary Fig. 6a, b), and intratumoral virus activity is strongly diminished within the first three days after treatment (Supplementary

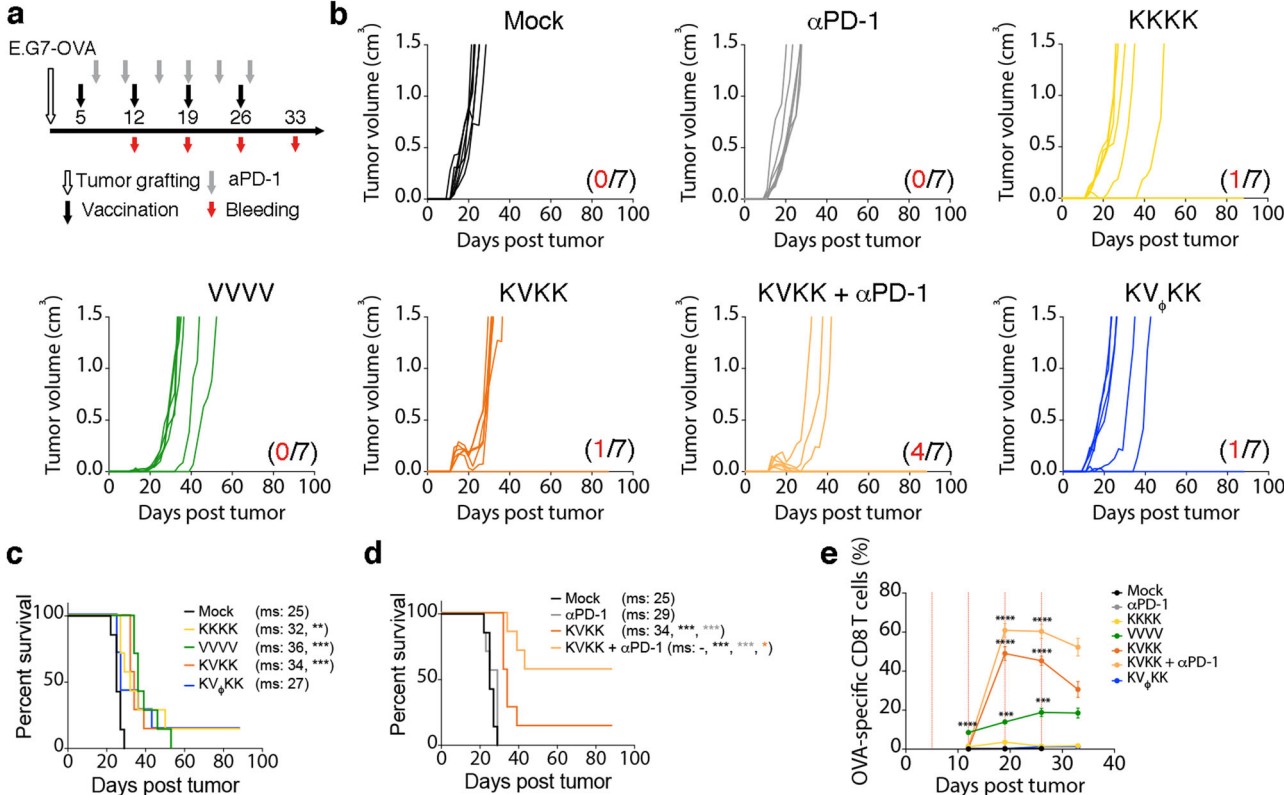

**Fig. 5 Therapeutic effect of heterologous vaccine in syngeneic tumor model expressing ovalbumin. a–e** C57BL/6J mice were injected s.c. with E.G7-OVA cells and vaccinated with KISIMA-OVA (K) s.c. or VSV-GP-OVA (V) i.v. on days indicated in schematic (**a**). 200 µg αPD-1 antibody was given i.v. twice weekly and blood was drawn for tetramer analysis as shown ($n = 7$). **b** Tumor growth and (**c**, **d**) survival is depicted with red numbers indicating long-term remissions within a group. Pairwise Log-rank test was performed (*$p < 0.05$; **$p < 0.01$; ***$p < 0.001$). **e** Frequency (mean ± SEM) of OVA-specific CD8+ T cells in circulation is shown. One-way ANOVA with Tukey's multiple comparison was performed and significance displayed only for each vaccinated group vs. mock (*$p < 0.05$; **$p < 0.01$; ***$p < 0.001$; ****$p < 0.0001$). The experiments (**b–d**) were repeated once with i.m. application of VSV-GP-OVA. Study (**e**) was performed once. Source data and p-values are provided in the Source Data File.

Fig. 6c). TC-1 tumors also do not respond to single VSV-GP treatment (Supplementary Fig. 6d, e) and VSV-GP replication in vivo is limited to an initial infection period (Supplementary Fig. 6f).

Therapeutic vaccination with both KISIMA-OVA and VSV-GP-OVA homologous as well as heterologous prime-boost as shown in Fig. 5a, significantly delayed growth of E.G7-OVA tumor cells, resulting in an increased median survival (Fig. 5b,c). Interestingly, a strong regression of large established tumors was only observed following boost in KVK vaccinated mice. The tumor regression correlated with the potent OVA-specific CD8+ T cells in the periphery (Fig. 5e, Supplementary Fig. 7e). In contrast, in MC-38 colorectal cancer model, in which the KISIMA-Mad24 homologous and heterologous prime-boost vaccinations (Fig. 6a) showed a low therapeutic efficacy, VSV-GP-Mad24 homologous vaccination was highly efficient, resulting in over 50% of complete tumor regression (Fig. 6b, c). This effect exemplifies the response to the oncolytic component of VSV-GP, as similar results were obtained using multiple VSV-GP ($V_\phi$) while the amplitude of the antigen-specific CD8+ T cells response did not correlate with antitumoral effect (Fig. 6e, Supplementary Fig. 7j). Results in TC-1 tumor model (Fig. 6f) were similar to the ones obtained in the oncolytically resistant E.G7-OVA model, with all vaccination schedules resulting in delayed tumor growth and increased median survival (Fig. 6g, h). Although frequency of HPV-specific CD8+ T cells in the periphery is higher in KVK-treated mice compared to homologous VSV-GP-HPV treated mice (Fig. 6j), they were similar in the tumor-infiltrating

leukocytes (Fig. 2b–e). This might explain why homologous VSV-GP-HPV was equally effective as heterologous prime-boost schedule despite lack of correlation between circulating HPV-E7-specific and tumor-size (Supplementary Fig. 7o). In order to address the role of a KISIMA-HPV prime, VSV-GP-HPV treatment at 14 days post tumor (time of boost) was assessed with or without KISIMA-HPV prime (Supplementary Fig. 8a). While virus treatment alone led to the slowing of tumor growth, no remission was observed (Supplementary Fig. 8a, c). In contrast, virus treatment following KISIMA-HPV prime led to a complete remission in all tumors; even in large tumors (Supplementary Fig 8b, c). This strongly indicates that priming with KISIMA-HPV is essential to induce tumor regression of sizeable tumors treated with virus two weeks after grafting. Whether this tumor antigen-specific KISIMA prime affects the intratumoral virus activity was assessed by daily measurements of virally encoded luciferase reporter gene activity (VSV-GP-Luc). Importantly, as the VSV-GP-Luc virus does not express the E7 tumor antigen cassette, any effects are predominantly based on tumor infection and lysis. Priming was performed at day 7 post tumor with vehicle, a non TC-1 related antigen prime (KISIMA-OVA prime), or the TC-1 tumor-specific prime (KISIMA-HPV prime). As shown above with VSV-GP, no effect on tumor growth kinetic was observed in any of the VSV-GP-Luc treated tumors ($10^8$ TCID$_{50}$ i.v.), compared to untreated mock control (Supplementary Fig. 9a). Importantly, daily bioluminescence measurements revealed no differences in antigen-specific, non-specific or buffer-prime intratumoral luciferase signals (Supplementary Fig. 9b–d).

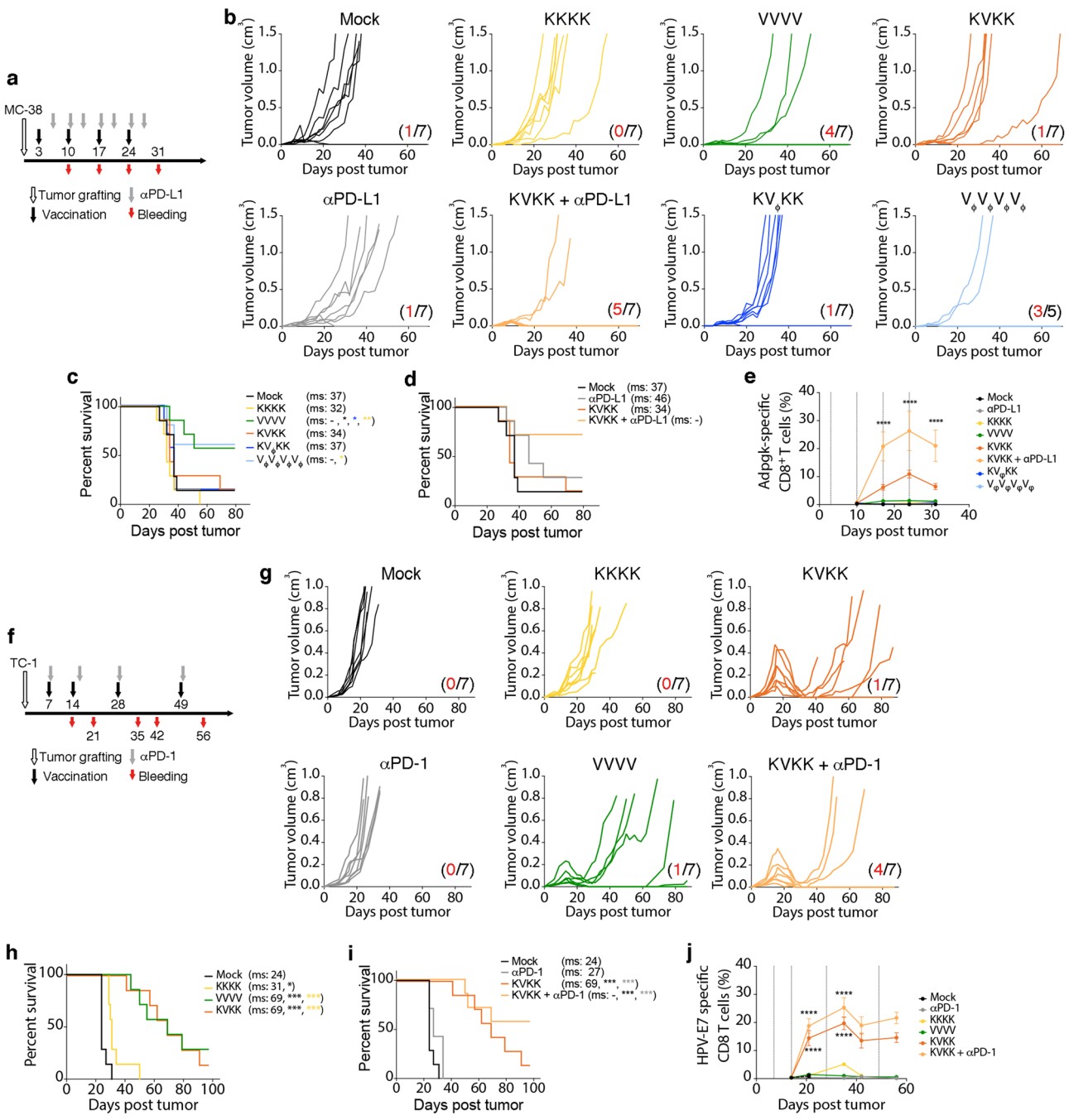

**Fig. 6 Efficacy of therapeutic cancer vaccination using KISIMA-TAA and VSV-GP-TAA in syngeneic tumor models targeting neo-epitopes and viral oncoprotein.** C57BL/6J mice were subcutaneously injected with (**a–e**) MC-38 cells or (**f–j**) TC-1 cells. Mice were immunized with (**a–e**) KISIMA-Mad24 or VSV-GP-Mad24 or (**f–j**) KISIMA-HPV or VSV-GP-HPV on days indicated in the schematic (**a**, **f**), respectively. Additionally, mice received 200 μg of (**a–e**) αPD-L1 antibody i.p. or (**f–j**) αPD-1 antibody i.v. as indicated in the schematic (**a**, **f**), respectively. **b**, **g** Tumor growth curves and (**c**, **d**, **h**, **i**) survival of the animals is depicted with red numbers indicating long-term remissions within a group. Pairwise Log-rank test was performed (*p < 0.05; **p < 0.01; ***p < 0.001). **e**, **j** The frequency of circulating (**e**) Adpgk-specific or (**j**) HPV-E7-specific CD8+ T cells is depicted as mean ± SEM. 5–7 mice analyzed per treatment group as indicated in (**b**, **g**). One-way ANOVA with Tukey's multiple comparison was performed and significance displayed only for each treatment group vs. mock (****p < 0.0001). Studies (**b–d**) have been repeated twice, once with PD-1 checkpoint combination treatment, once without checkpoint combination. T cell analysis in e has been repeated once. Studies (**g–i**) were repeated independently in three additional experiments, including once with PD-1 checkpoint combination. T cell analysis in (**j**) has been repeated once. Source data and p-values are provided in the Source Data File.

Together, these data suggest that KISIMA-HPV prime lays the immunological foundation for the strong tumor remission that follows VSV-GP-HPV boost in the TC-1 tumor model.

Despite KVK heterologous prime-boost inducing significant tumor remission in both E.G7-OVA and TC-1 tumor models, tumors were able to relapse at later time points, despite the presence of high levels of circulating antigen-specific CD8+ T cells. In order to understand the underlying mechanism of tumor escape, relapsing TC-1 tumors were harvested on day 42 post-implantation (Supplementary fig. 10a) for in-depth characterization of antigen-specific TILs. Similar to the analysis performed 7 days after the first boost (Fig. 2, Supplementary

Fig. 3), both HPV-E7- (Supplementary Fig. 10b–e) and VSV-N- (Supplementary Fig. 10f–i) specific CD8+ T cells were more abundant within the tumor (Supplementary Fig. 10d, e, h, i) compared to the periphery (Supplementary Fig. 10b, c, f, g), irrespective of the vaccine regimen. While the frequency of intratumoral HPV-E7-specific CD8+ T cells was similar to the one at the earlier time point, a higher proportion of these cells co-expressed PD-1 and Tim-3 exhaustion markers (Supplementary Fig. 10l). Further, the functionality of HPV-E7-specific CD8+ T cells was strongly reduced, with only a small proportion of cells still producing IFN-γ and TNF-α or expressing the CD107a degranulation marker (Fig. 2d, Supplementary Fig. 10n). In contrast, peripheral HPV-specific CD8+ T cells (Supplementary Fig. 10j) and antiviral CD8+ T cells (Supplementary Fig. 10k, m) did not progressively upregulate Tim-3 at the time of tumor relapse.

The transcriptome of TC-1 tumors undergoing relapse was compared to that of TC-1 tumors responding to therapeutic vaccination. Principal component analysis showed that respond-ing tumors from KV vaccinated mice clustered together and had a distinct gene expression pattern compared to the other samples (Supplementary data 1 and Supplementary Fig. 11). Notably, the relapsing tumors both from VV and KV immunized groups clustered closer to untreated and KK treated tumors, suggesting loss of immune activation. This was further confirmed as few panel genes were upregulated (Supplementary data 3 and Supplementary Fig. 12a) and most genes were downregulated (Supplementary data 3 and Supplementary Fig. 12b) in relapsing tumors when compared to responding tumors from the same treatment group. This was also reflected in the overall reduction in gene signatures associated with cytotoxic T cell infiltration (Supplementary Fig. 12c), DC function (Supplementary Fig. 12e) and loss in antigen presentation (Supplementary Fig. 12g). Additionally, cytokines (Supplementary Fig. 12d) and chemo-kines (Supplementary Fig. 12f) necessary for attracting T cells and other immune cells into the tumors were drastically decreased in relapsing tumors. Together, these data highlight that tumor relapses are linked to an unsustained and fading immune activation, which in turn could be addressed by concomitant application of immune checkpoint inhibitor (CPI) compounds.

Therefore, the combination of KVK heterologous prime-boost with checkpoint blocking antibodies was assessed in order to limit intratumoral T cell exhaustion and avoid tumor relapses. Strong synergy was observed between checkpoint blockade and hetero-logous vaccination in all 3 tumor models (Figs. 5, 6). PD-1 blockade alone had no effect on tumor growth or median survival of mice bearing either E.G7-OVA (Fig. 5b, d) or TC-1 tumors (Fig. 6g, i). However, when combined with heterologous vaccination, αPD-1 treatment prevented tumor relapse after complete regression, resulting in a high number of long-term survivors while median survival was not reached (Figs. 5d, 6i). In contrast, αPD-L1 antibody monotherapy delayed tumor growth in MC-38 bearing mice, which may be due to high mutational burden and presence of endogenous tumor-reactive T cells. However, in combination with KVK heterologous vaccination, αPD-L1 treatment strongly increased vaccine efficacy, resulting in long-lasting complete regression in over 70% of animals (Fig. 6b,d). Additionally, combination with CPI also promoted expansion of vaccine induced OVA- (Fig. 5e, Supplementary Fig. 7a), Adpgk- (Fig. 6e, Supplementary Fig. 7f) and HPV-E7-specific (Fig. 6j, Supplementary Fig. 7k) CD8+ T cells in the periphery. Surprisingly, circulating antiviral CD8+ T cells were unaffected by checkpoint inhibition in all three tumor models (Supplementary Fig. 7b,c,g,h,l,m) but the ratio of antitumor to antiviral CD8+ T cells in circulation was not greatly enhanced by combining checkpoint blockade antibodies with KVK vaccination

(Supplementary fig 7d, i, n). Interestingly, αPD-1 therapy did not lead to further expansion of circulating OVA-specific CD8+ T cells in mice bearing B16-OVA tumors when combined with heterologous vaccination (Supplementary Fig. 13d). This might explain why αPD-1 therapy failed to further enhance the therapeutic effect of KVK vaccination (Supplementary Fig. 13a) in this tumor model, reflected in delayed tumor growth (Supplementary Fig. 13b) and prolonged survival (Supplementary Fig. 13c).

Tumor re-challenge of long-term survivors was performed in E.G7-OVA, MC-38 and TC-1 tumor models. KVK heterologous prime-boost (+/- CPI) developed an effective memory response, as almost all the re-challenged mice rapidly rejected the newly implanted tumor (Supplementary Table 3–5). Interestingly, in TC-1 bearing mice, only 60% of homologous VSV-GP-HPV treated long-term survivors were protected against re-challenge, possibly reflecting the reduced formation of memory precursor cells compared to the heterologous vaccination. Similarly, only 75% of long-term survivors which had successfully rejected MC-38 tumors upon homologous VSV-GP-Mad24 vaccination remained tumor-free after rechallenge.

Taken together, the data strongly support the combined application of KISIMA-TAA cancer vaccine and VSV-GP-TAA oncolytic vaccine in a heterologous prime-boost regimen. This approach leads not only to significantly enhanced peripheral and intratumoral T cell levels, but also to a profound reshaping of the TME towards a more immune-supportive composition.

## Discussion

Tumor-targeting vaccines have long been one of the pillars of cancer immunotherapy developments and have regained increased momentum in recent years[30]. Major challenges still exist, such as inducing tumor-specific T cells with high frequency and good quality, and overcoming the immune suppression within the TME[31]. In the present study, a heterologous prime-boost combination was presented involving KISIMA-TAA, a highly potent self-adjuvanting protein vaccine, with VSV-GP-TAA, a vaccine based on an oncolytic virus platform. Compared to the respective homologous vaccine regimen, the KV combi-nation promoted large quantities of polyfunctional antigen-specific T cells, their enhanced infiltration into the tumor, as well as a deep remodeling of the TME, essentially addressing these two vaccine limitations. Furthermore, the vaccine approach rendered either resistant tumor models susceptible to CPI treatment, or enhanced existing CPI responses.

Including oncolytic viruses to cancer vaccine regimens can extend the therapeutic modality by direct tumor lysis and induction of immunogenic cell death (ICD)[22]. However, many tumors develop antiviral mechanisms limiting viral propagation[8]. Therefore, we included tumor models with varying OV resistance. E.G7-OVA tumors were completely resistant to VSV-GP. TC-1 and B16-OVA tumors were permissive in vitro but restricted through IFN-mediated antiviral control, resulting in restricted viral propagation in vivo, similar to previous findings[16,20]. MC-38 tumors were highly permissive in vitro without IFN protection. Surprisingly, viral propagation and lytic therapy after single application was limited in vivo. Hence, the tumor models studied show suboptimal responses to oncolytic monotherapy. This contrasts previous studies with heterologous vaccine regimen with systemic virus application and oncolytic efficacy, although no long-term data were presented[32].

The KV combination generated the strongest CD8+ T cell responses when VSV-GP-TAA was applied as a boost compared to a VSV-GP prime. This is consistent with the boosting potential of VSV shown in previous studies using various priming

partners[23,26,33]. Importantly, the combination of tumor-specific prime with non-antigen-encoding VSV-GP showed no therapeutic effect in the current study, confirming previous reports on the importance of the coordinated TAA expression[32]. Although homologous VSV-GP-TAA also showed efficacy in some models, tumor control was limited. The combination with KISIMA-TAA prime strongly enhanced the proportion of memory T cells, and tumor relapse was delayed. An additional key finding of our heterologous combination was the increase in polyfunctional CD8+ T cells, including IFN-γ, TNF-α, and CD107a as shown with ex vivo intracellular staining. Importantly, the transcriptome analysis of freshly isolated tumor tissue corroborates the notion that heterologous KV vaccination profoundly enhances T cell activity and functionality.

Whereas KISIMA has routinely been administrated s.c.[4], oncolytic VSV-GP has been extensively tested in i.v. and intratumoral applications[16,19,34] and the main route for prophylactic VSV-GP based vaccines has been i.m.[18]. In this study systemic VSV-GP-TAA application was shown to significantly enhance the CD8+ T cell responses, without inducing any signs of toxicity. This is in line with previous studies that showed strong adaptive immune responses to VSV variants associated with strong splenic uptake. On the other hand, i.v. application of VSV results in rapid clearance of non-cell bound free virus from the circulation within a few minutes[35]. Nonetheless, systemic injection of cancer vaccines has recently been used including in clinical trials[33,36], offering for oncolytic virotherapy the perspective to facilitate seeding in disseminated tumor masses.

The detection of OVs by the immune system and the resulting mounting of antiviral effects also harbor therapeutic benefits[37]. Viral infection of tumor tissue results in a rapid innate response, driving a proinflammatory environment with subsequent infiltration of antigen presenting cells[10]. However, the adaptive antiviral response can interfere with the induction of the target antigen-directed T cell population and heterologous combinations of vaccine partners may ameliorate this imbalance[22,23]. Although VSV-GP was previously shown to curb induction of neutralizing anti-vector antibodies in mouse models, allowing repetitive application of the same virus[18], potent T cell responses are triggered against viral epitopes[34]. We monitored post prime anti-VSV-N responses of around 30% of the circulating CD8+ pool, independent of whether the TAA was OVA, E7, or Adpgk and also detached from the permissivity status of a tumor. Anti-vector CD8+ T cell responses were then reduced in heterologous combinations for both OVA and E7 antigens after a second boost with KISIMA-TAA. Similarly, the anti-VSV-N response was also shown to be significantly reduced in another heterologous prime-boost study using the human dopachrome tautomerase (hDCT) as TAA[23]. While the dynamic of the antiviral T cell response might vary depending on the vaccine construct and target antigen, the ratio of antitumor versus antiviral response was consistently and profoundly shifted towards the antitumor response, confirming the basic rationale of heterologous cancer vaccines[22]. Although tumor bystander killing by antiviral T cells via cytokine release has been reported[37], this effect might be reduced in our study as the selected tumor models had limited or excluded permissivity for VSV-GP propagation. Finally, while CPI treatment enhanced the TAA directed T cell response for all three tumor antigen models, frequencies of antiviral T cells were not affected in contrast to ongoing discussions on CPI and OV combination and the role of antiviral immunity therein[38].

KV heterologous vaccine not only leads to a dramatic increase in T cell infiltration but also induces a profound remodeling of the immunosuppressive TME. The enhanced T cell infiltration can be attributed to potent antitumor T cell response[4,36] and VSV-GP-HPV mediated T cell trafficking as has been described for VSV-GP[16] and other OVs as monotherapy[39]. Among the various cytokines upregulated in the tumor or plasma after KV vaccination, several are secreted by DCs in response to the KISIMA vaccine[4] or are produced in permissive tumors after oncolytic VSV-GP therapy[16]. While CCL2, CCL3, CCL4 and CCL5 can directly promote T cell recruitment into the tumor[40], CCL3, CCL4 and CCL5 also enhance recruitment of cross-presenting DCs[41] and M1-like TAMs[42] capable of producing CXCL9 and CXCL10[41,42], which further increases T cell infiltration in the tumor[40,43] as seen in TC-1 tumors after KV vaccination. The upregulation of type I and type II IFNs observed after heterologous prime-boost could explain the enhanced antigen presentation[44], which would lead to improved recognition and killing of tumor cells. The strong upregulation of IFN-γ and the influx of CD4+ effector T cells as result of KV vaccination might contribute to repolarization of M2-like TAMs as antigen-specific Th1 cells can reprogram TAM-2[45]. Since both KISIMA-HPV and VSV-GP-HPV contain the CD4 epitopes, the role of tumor-specific CD4+ T cells might be interesting to further investigate in the future.

TC-1 and E.G7-OVA tumors, both initially resistant to checkpoint blockade[4], became susceptible to anti-PD-1 treatment consistent with previous KISIMA studies[4] or RNA vaccines[36]. In contrast, MC-38 tumors expressing many neo-epitopes[28,46] are sensitive to inhibition of PD-1/PD-L1 interaction[4]. However, combining αPD-L1 antibody with KV vaccination did increase the number of long-term survivors. Importantly, clonal expansion of tumor-specific T cells in the periphery was observed upon checkpoint blockade which has been described in clinical studies[47]. Surprisingly, virus-specific T cells did not proliferate after checkpoint blockade, despite expressing PD-1 and Tim-3, which might have a positive effect on the balance of antiviral versus antitumoral CD8+ T cells. The KV heterologous prime-boost approach holds great promise for patients with primary or acquired resistance to CPI[48] due to its ability to induce tumor-specific T cell, improve T cell infiltration and increase tumor inflammation, even in tumors with limited permissivity for the oncolytic virus. With the KISIMA platform currently undergoing Phase 1 testing and VSV-GP having undergone an extensive preclinical safety program, clinical explorations of the combined KV regimen shall begin soon.

## Methods

**Ethics statement**. All animal experiments were approved by the Institutional Animal Care and Use Committee (ZVTA) from the Medical University Innsbruck and the Austrian Federal Ministry of Science, Research and Economy (BMBWF) in accordance with the "Tierversuchsgesetz 2012" (BGBl, I Nr 114/2012) and by institutional and cantonal Geneva veterinary authorities in accordance with Swiss Federal law on animal protection and performed according to institutional guidelines of the Medical University of Innsbruck, Austria and of Swiss Federal law on animal protection, respectively. All animals were housed in a BL2 facility in individually ventilated cages with a 12-h light/dark cycle with unrestricted access to food and water. Temperature in animal facilities was 20–24 °C and humidity was 55 ± 10%.

**Tumor cell lines for implantation**. E.G7-OVA cells were purchased from ATCC (Manassas, Virginia, US) and maintained in complete RPMI 1640 medium with 0.4 mg/ml geneticin (Life Technologies, Carlsbad, California, US). B16-OVA cells were provided by Bertrand Huard (University of Grenoble-Alpes, Grenoble, France) and maintained in complete RPMI 1640 medium with 1 mg/ml geneticin. TC-1 cells were provided by T.C. Wu (Johns Hopkins University, Maryland, US) and cultured in complete RPMI 1640 with 0.4 mg/ml geneticin. MC-38 cells were a kind gift from Gottfried Baier (Medical University of Innsbruck, Austria) and were maintained in complete DMEM containing 5% gentamicin. None of the cell lines used in this study were commonly misidentified lines listed in ICLAC. Six- to eight-week-old female C57BL/6RJ or B6(C)/Rj-Tyr$^{c/c}$ mice were obtained from Janvier (Le Genest St Isle, France) or Charles River (C57BL/6RJ stock number 632; L'Arbresles, France). For tumor implantation, mice were injected subcutaneously with $3 \times 10^5$ E.G7-OVA cells, $2 \times 10^5$ B16-OVA or $2 \times 10^5$ MC-38 cells in the right flank or with $1 \times 10^5$ TC-1 cells in the back. For monitoring tumor growth, tumor diameter was measured 2–3 times per week using a caliper and volume was calculated using the formula: $0.4 \times \text{length} \times \text{width}^2$. Mice were euthanized when tumor-size reached 1 or 1.5 cm$^3$ depending on the respective institutional veterinary authorities or tumors showed signs of ulcerations. Animals were euthanized by $CO_2$ asphyxiation and cervical dislocation.

**Generation of vaccine constructs**. Recombinant protein vaccine KISIMA constructs were designed in-house and produced in *E. coli* by Genscript (Piscataway, New Jersey, US). During purification process, endotoxins were removed from vaccines through extensive washes with Triton-X114 followed by subsequent affinity chromatography. Only the batches with endotoxin level below 10 EU/mg protein (LAL chromogenic assay) were used. KISIMA-OVA vaccine contains both CD8 and CD4 H-2$^b$ epitopes from Ovalbumin whereas KISIMA-Mad24 (multi-antigenic domain 24) contains the immunogenic neo-epitopes Adpgk and Reps1. KISIMA-HPV contains both CD8 and CD4 H-2$^b$ epitopes from E7 HPV.

Recombinant viruses VSV-GP, VSV-GP-OVA and VSV-GP-Luciferase (VSV-GP-Luc) have been described previously[18,21] whereas VSV-GP-Mad24 and VSV-GP-HPV were generated de novo. VSV-GP-Mad24 (multi-antigenic domain 24) expresses the immunogenic neo-epitopes Adpgk and Reps1[28] and VSV-GP-HPV encodes the attenuated E6/E7 fusion construct[49] in addition to wild type E2. All the recombinant virus variants were recovered using a helper virus-free calcium phosphate transfection in 293T cells with expression plasmids of T7 polymerase (10 μg), VSV proteins N (2.8 μg), P (1.8 μg) and L (0.6 μg) together with the respective vaccine construct containing VSV-GP vector (10 μg)[50]. After detection of cytopathic effects and expansion on BHK-21 cells, virus progeny were plaque-purified twice and amplified on BHK-21 cells. A 0.45 μm filtration step was followed by sucrose cushion (20%) centrifugation. Viruses were titrated via TCID$_{50}$ assay.

**Immunization and checkpoint blockade**. Vaccination regimens were based on previously published homologous KISIMA vaccinations studies[4]. For the E.G7-OVA and the MC-38 tumor models, mice received the first vaccination on day 5 and day 3 post tumor implantation, respectively. This was followed by 3 boost immunizations at 7 days interval. Mice bearing TC-1 tumors were grouped prior to first immunization on day 7 post tumor implantation in order to start with comparable mean tumor size for each treatment group. Vaccination was repeated on days 14, 28 and 49 after tumor implantation. Mice were vaccinated either with 2 nmol recombinant protein vaccine KISIMA-TAA (targeting the relevant TAA) administered s.c. at the tail base or with 1×10$^7$ TCID$_{50}$ of the respective VSV-GP-TAA or VSV-GP injected i.v. into a lateral tail vein on days indicated above. Unless otherwise noted, vaccine treatment intervals in tumor-bearing mice followed the application regimen from immunogenicity studies in non-tumor-bearing mice for the respective vaccine combinations.

For checkpoint blockade, E.G7-OVA tumor-bearing mice received 200 μg αPD-1 antibody (clone RMP1-14, BioXcell, Lebanon, New Hampshire, US) intravenously every 4 days starting on day 7 post tumor implantation. For the MC-38 tumor model, 200 μg of αPD-L1 antibody (clone 10 F.9G2, BioXCell) was injected intraperitoneally on days 6, 10, 13, 17, 20, 24 and 27 post tumor implantation. Mice bearing TC-1 tumors received intravenous injections of 200 μg αPD-1 antibody 7, 15, 28 and 49 days after tumor implantation.

**Flow cytometry**. Single cell suspensions were prepared from spleen and bone marrow by mechanical dissociation using a 40 μM cell strainer. This was followed by lysis of erythrocyte using Pharm Lyse$^{TM}$ Lysing buffer (BD Biosciences, San Jose, California, US). For whole blood, lysis was carried out after surface staining. Tumor-infiltrating leukocytes (TILs) were purified using mouse tumor dissociation kit (Miltenyi) and Gentle MACS with heating system (Miltenyi Biotec, Bergisch Gladbach, Germany), following manufacturer instruction. Single cell suspension was obtained after filtration through a 70 μm cell strainer and CD45+ cells were purified using CD45 TIL microbeads (Miltenyi Biotec) following manufacturer protocol and used for flow cytometry analysis.

For the detection of antigen-specific CD8+ T cells, whole blood or single cell suspensions from spleen, bone marrow or tumors were labeled with fluorescently-labeled peptide-MHC multimers as listed in Supplementary Table 6. This was followed by surface staining with antibodies listed in Supplementary Table 7. Dead cells were labeled using LIVE/DEAD™ Fixable Near-IR Dead Cell Stain Kit (Thermo Fischer Scientific, Waltham, MA, US). For phenotyping tumor-infiltrating leukocytes subsets, the monoclonal antibodies (mAbs) listed in Supplementary Table 8 were used. Dead cells were identified with LIVE/DEAD yellow or aqua fluorescent reactive dye from Life Technologies and were excluded from analyses. Intracellular staining was performed after stimulation with the indicated peptides and in presence of CD107a mAb for 6 h in the presence of Brefeldin A (GolgiPlug, BD Biosciences). Intracellular staining was done with mAbs to IFN-γ, TNF-α, and corresponding isotype controls as listed in Supplementary Table 9. For granzyme B intracellular staining, cells were cultured for 4 h in the presence of Brefeldin A (GolgiPlug, BD Biosciences). Intracellular staining was done with mAb to granzyme B (Table S9). Fixation and permeabilization was carried out using the BD Bioscience kit according to manufacturer's instructions. Samples were acquired on FACS Canto II (BD Biosciences), Gallios flow cytometer (Beckman Coulter), or Attune (ThermoFisher).

Flow cytometry data was analyzed with FlowJo software version 10.5.3 (FlowJo, LLC, Oregon, US) or Kaluza (Beckman Coulter) software.

**Bioimaging**. For monitoring intratumoral viral replication in vivo, C57BL/6Rj or B6(C)/Rj-Tyr$^{c/c}$ Albino mice bearing subcutaneous tumors were treated intratumorally

or intravenously with indicated dose of VSV-GP-Luciferase (VSV-GP-Luc). For luciferase imaging, 1.5 mg D-luciferin (Promega, Madison, WI, USA) was administered intraperitoneally 15 m prior to measurements using the Lumina system (IVIS Lumina II, Perkin Elmer, Waltham, Massachusetts, USA)[16]. Caliper Live Sciences-Living Image software (4.3.1.) was used for data acquisition and analysis

**Cell viability and IFN-I resistance assay**. Cells (2×10$^4$ per well) were seeded in 96-well plates and treated with universal IFN-α A/D (PBL Assay Science, Piscataway, New Jersey, US) at different concentrations for 16 h followed by infection with different doses of VSV-GP virus as indicated. Seventy-two hours later thiazolyl blue tetrazolium bromide (MTT) was added. After 4 h, cells were dissolved in 0.1 M HCl with 1% SDS and after another 4 h colorimetric changes were quantified at 540 nm[16].

**Transcriptome analysis**. Snap frozen tumor tissue was homogenized using RLT buffer (Qiagen, Hilden, Germany) and SpeedMill PLUS (Analytik Jena, Jena, Germany) followed by Phenol/Chloroform extraction. RNA was isolated from the aqueous phase using RNeasy Mini kit (Qiagen) according to manufacturer's instructions and total RNA was used for differential expression analysis using the nCounter PanCancer Immune Profiling Panel and the nCounter FLEX Analysis System (NanoString Technologies, Seattle, WA, USA). Profiled data were pre-processed following manufacturer's recommendations[51] and heatmaps were generated using nSolver 4.0 software. Normalized gene counts from nSolver software were used to calculate the principal component analysis (PCA) using ClustVis[52]. Venn diagrams were generated using the webtool (http://bioinformatics.psb.ugent.be/webtools/Venn/).

**Multiplex ELISA**. Plasma cytokines and chemokines were analyzed using LEGENDplex$^{TM}$ Mouse Anti-Virus Response Panel (13-plex) (BioLegend, San Diego, California, US), according to manufacturer's instructions. Data were analyzed using LEGENDplex™ Cloud-based Data Analysis Software (BioLegend).

**Immunohistochemistry**. Tumors were fixed in 4% buffered formaldehyde solution and embedded in paraffin. 2–3 μm thick sections were stained with hematoxylin and eosin (HE). Immunohistochemistry (IHC) was used to assess T -cells using the primary monoclonal antibody (clone D4W2Z, dilution 1:2000) against CD8α (Cell Signaling, Danvers, Massachusetts, US). For antigen retrieval sections were heated in citrate buffer. The following steps were performed either manually or automatically in an autostainer (Lab Vision AS 360, Thermo Scientific, Freemont, USA): Blocking of endogenous peroxidase by incubation in H$_2$O$_2$, reducing background by application of a protein blocking reagent and applying the respective primary antibody. A secondary antibody formulation conjugated to an enzyme-labeled polymer and Di-amino-benzidine as chromogen were used. Sections were counterstained with hematoxylin. An experienced pathologist blinded to treatment regimens evaluated sections with an Olympus BX-53 microscope (Olympus, Tokyo, Japan).

**Figure generation and statistics**. Image processing and figure composition was done using Adobe Photoshop CS6 (Adobe, Mountain View, California, US). Statistical analyses were performed using Prism software (GraphPad) and considered statistically significant if $P < 0.05$. Used statistical tests included unpaired 2-tailed t test, one-way ANOVA with Tukey's multiple comparison, two-way ANOVA test with Sidak's multiple comparison, Kruskal–Wallis test, Mann-Whitney test, and Log-rank test as indicated in the figure legend. Benjamini–Yekutieli procedure was used to calculate the FDR from the *p*-values returned by the t-test.

**Reporting summary**. Further information on research design is available in the Nature Research Reporting Summary linked to this article.

## Data availability

The authors confirm that the pertinent data supporting the findings of this study are included in the article and the supplementary materials. The gene expression data of Nanostring nCounter is provided as supplementary data file. Source data is provided as a source data file. Source data are provided with this paper.

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

## Acknowledgements

The authors thank E. Richter, J. Schoergenhuber, N. Payr, D. Berger (Christian Doppler Laboratory for Viral Immunotherapy of Cancer), S. Carboni, W. Di Berardino-Besson (AMAL Therapeutics), and P. Kodajova (University of Veterinary Medicine Vienna) for excellent technical help. We also thank M. Docquier and D. Chollet (iGE3 Genomics Platform, University of Geneva) for their support for the Nanostring® assay. This research was funded by the Christian Doppler Research Association and the Austrian Research Promotion Agency (FFG Bridge Project # 877143).

## Author contributions

K.D., E.B., and M.R. designed and performed experiments, analyzed and interpreted data, and wrote and edited the manuscript. T.H., S.D., L.M.S., and K.A. performed experiments and analyzed data. T.N., B.Sp. and J.K. generated the viruses. S.H. and L.K. performed and analyzed IHC assay. D.V.L., K.E., and K.D. conceived the project. M.D. and G.W. conceived the project, supervised the research, wrote and edited the manuscript.

## Competing interests

D.v.L. is inventor on a patent related to VSV-GP. D.v.L. and G.W. serve as scientific advisors for Boehringer Ingelheim GmbH. T.N., B.Sp. and K.E. are employees of ViraTherapeutics GmbH and Boehringer Ingelheim International GmbH. M.D. and E.B. are inventors on patents related to KISIMA vaccine platform. M.D., E.B., and M.R. are employees of AMAL Therapeutics and Boehringer Ingelheim International GmbH. The other authors declare no competing interest.
