## [Peer Review File · Nature Communications]

REVIEWER COMMENTS

Reviewer #1 (Remarks to the Author):

This manuscript by Das et al. describes dual treatment of cancers with KISMA (a multi-domain self-adjuvating vaccine) and VSV-GP (a modified oncolytic vesicular stomatitis virus expressing LCMV-GP) in increasing T cell specificity, increased cytokine production, tumor regression, and development of long-term immunity. While the experiments show impressive tumor rejections and increase in immunity, some major and minor concerns exist:

Major Concerns:

1. Timelines, modes of treatment, and administration numbers vary throughout the studies. For example, some non-tumor studies use a Day 0 prime – Day 7 boost – Day 14 boost cycle with VSV given i.v. and KISMA s.c. with 3 total treatments. TC-1 tumor models used a Day 0 tumor graft – Day 7 prime – Day 14 boost model with 2 total treatments. OVA, MC-38, and subsequent TC-1 tumor models use a Day 0 tumor – Day 5 prime - aPD-1 twice – Day 7 boost -aPD-1 twice...repeat, -- with variance between each cell line -- and VSV given by i.t at times instead of i.v. What is the rationale for these changes between experiments and possible effect on outcomes?

2. CD8+ T cell functionality is assessed by ex vivo restimulation of cells. Unfortunately, this afford cells (even functionally exhausted ones) an opportunity for additional synthetic re-stimulation after which they can produce cytokines ex vivo that would otherwise not be produced in vivo (particularly within a suppressive tumor microenvironment). Steps should be taken to assess actual capabilities of these cells (by staining for markers which do not require ex vivo restimulation, such as granzyme B and often IFN-gamma; or by direct analysis after tissue dissection/cell dissociation from animals that received protein export blockade [golgiplug/golgistop] in vivo a few hours prior to euthanasia).

3. There appear to be some inconsistencies amongst the data that need further investigation and/or consideration. It is mentioned that reduction in viral-specific T cells occurs (in a non-tumor model when compared to OVA) when either KVV or KVK are used (SFig 2i-k). However, in a tumor model (SFig 3b) KV priming appears to increase virus-specific T cells over VV. Also, in Figure 4c it appears that VV dramatically increases T cell infiltrates over KV in flow data, which is shown to be the opposite in 4d histology.

4. It is not clear (here or in the literature) that reducing anti-viral immunity necessarily results in a net positive effect of tumor reduction. Studies have shown previously that increased viral specific T cells in a tumor microenvironment can positively influence tumor growth. While VSV is oncolytic, studies have also shown that a factor in its anti-tumor ability is its immune recruitment. This should be considered and discussed further. (see: Dai et al, Science Immunology, 2017: <https://immunology.sciencemag.org/content/2/11/eaal1713.long>, Newman et al, PNAS 2019. <https://www.pnas.org/content/117/2/1119> Nelson et al, Cell Reports: <https://www.ncbi.nlm.nih.gov/pmc/articles/PMC6874401/>

5. Statistical analyses appear to be missing in some panels of figures (including but not limited to Figs 1a, 4b, S1ab, S2ik, S5, S6, S7 some panels, S9d).

6. It is unclear whether data shown are only from one experiment (with the n number of mouse listed) or if they are representative of more than one experiment conducted. Multiple experiments would be important for demonstration of reproducibility.

7. Many of the treatment regimens appear to start before tumors are well established, and a prophylactic rationale for such studies is unclear .

Minor Concerns:

1. All terms should be clearly defined (at least at first use). For example, it is unclear without

further searches what antigen Adpgk is, or what type of cell line TC-1 is.

2. For some figure panels, the groups being compared are unclear (including but not limited to Figs 2f, 2g, S3f,g)

Reviewer #2 (Remarks to the Author):

This well-written manuscript from Das et al reports on a series of preclinical studies that builds on their prior work with a novel chimeric protein cancer vaccine (KISIMA) combined with systemic administration of an oncolytic virus vaccine that encodes the same antigens, in a prime:boost approach. The KISIMA vaccine incorporates a multi-antigenic domain (Mad) with epitopes for CD8 and CD4 T cells + cell penetrating peptide (CPP) + peptide agonist for TLR2 and TLR4 (TLRag). This is abbreviated as K. The viral construct is built on a vesicular stomatitis virus (VSV) variant encoding tumor antigen, and abbreviated as V. The investigators report on extensive preclinical studies in 3 different tumor models, using model tumor antigens, viral tumor antigens, and mutated cancer neoantigens. The prime:boost strategy with KISIMA prior to oncolytic virus (KV), especially with followed with K boost (KVK) induces strong and durable T cell responses in circulation and infiltration of those cells in the tumors. T cell function status is assessed by flow cytometry and gene expression (Nanostring) and reveals both effector and memory T cells, and suggests less exhausted T cells infiltrating tumors when using KVK than other strategies. The tumor control with KVK vaccines is disappointing (1 of 7 tumors with durable control in some experiments), but addition of PD1 blockade (which is ineffective as monotherapy in the models tested) induces durable tumor control in about half of the mice. They also show that IV administration of V is more effective than IM, which is another novel finding. The manuscript includes extensive data in the main figures as well as in supplemental material. The methods are sound and are well-controlled and comprehensive. The work has strong implications for design of optimal cancer vaccines in a range of cancers. There is strong translational potential and high impact across the field of cancer immunology and immunotherapy. The statistical analyses are appropriate and rigorous.

The only criticism is in Figure 4, where the immunohistology images (4d) show higher CD8 densities with KV than with VV, but the data in 4b and 4c suggest that the CD8 infiltrates are likely greater on average with VV. It would be good to modify so that the images in 4d are more representative of the summary data in 4b and 4c, or to explain why there may be a discrepancy.

One additional question is whether the investigators have compared intratumoral injection of V instead of the IV approach that they have used. In humans, oncolytic viruses have mostly been administered intratumorally. Being able to induce benefit with IV administration is appealing, but it may be good to comment on this in the manuscript and to highlight the advantages of being able to administer IV (for tumors that are not readily accessible percutaneously).

Reviewer #3 (Remarks to the Author):

1. Das et al. present that the self-adjuvanting cancer vaccine combined with an oncolytic vaccine induces potent antitumor immunity. The authors also show that heterologous vaccination with KISIMA and VSV-GP-TAA could sensitize non-inflamed tumors to checkpoint blockade therapy. The concept of this experiment and the approach for the procedures are designed well on the whole.

2. The concept of self-adjuvanting cancer vaccine combined with an oncolytic vaccine does not seem to be novel. Can they emphasize what is novel about this study in Introduction or Discussion?

3. The order between KISIMA-TAA and VSV-GP-TAA as a priming is important?

4. The authors showed that a single intravenous (i.v.) administration of VSV-GP-OVA induced the highest frequency of antigen-specific CD8+T cell response compared to intraperitoneal (i.p.), subcutaneous (s.c.) or intramuscular (i.m.) injection. However, the i.v. administration only be effective because of its high blood levels.

Referee response letter

We would like to thank the reviewers for the very helpful and constructive suggestions. We have thoroughly revised the manuscript, added new data where required and discussed pertinent issues that lacked clarity. A point-by-point response to all of the referee's comments is listed below.

The response letter has been structured the following way:

Reviewers comments in grey

Responses in indented text

Revised manuscript text passages in italics

Reviewer #1 (Remarks to the Author):

This manuscript by Das et al. describes dual treatment of cancers with KISMA (a multi-domain self-adjuvating vaccine) and VSV-GP (a modified oncolytic vesicular stomatitis virus expressing LCMV-GP) in increasing T cell specificity, increased cytokine production, tumor regression, and development of long-term immunity. While the experiments show impressive tumor rejections and increase in immunity, some major and minor concerns exist:

We appreciate the constructive and insightful analysis of the reviewer. We have expanded the analysis of some included studies and added new results as well. All comments and suggestions of this reviewer were addressed, as outlined below:

Major Concerns:

1.

Timelines, modes of treatment, and administration numbers vary throughout the studies. For example, some non-tumor studies use a Day 0 prime – Day 7 boost – Day 14 boost cycle with VSV given i.v. and KISMA s.c. with 3 total treatments. TC-1 tumor models used a Day 0 tumor graft – Day 7 prime – Day 14 boost model with 2 total treatments. OVA, MC-38, and subsequent TC-1 tumor models use a Day 0 tumor – Day 5 prime - aPD-1 twice – Day 7 boost -aPD-1 twice...repeat, -- with variance between each cell line -- and VSV given by i.t at times instead of i.v. What is the rationale for these changes between experiments and possible effect on outcomes?

We appreciate this comment and have added additional clarification to the manuscript. What may look like a very heterogenous and arbitrary interval setting between treatments actually follows a rather systematic approach. In the following paragraphs, we explain in detail the rationale behind the treatment regimen used. The administration schedule for the vaccination and checkpoint blockade was based on previous data, depending on the growth kinetics of the respective tumor

model. The initial KISIMA vaccination schedule has been established in tumor-free animals, based on the frequency and the quality of the T cells response along with memory induction, as previously published (Belnoue E et al., JCI Insight. 2019 - cited in the manuscript). This schedule with 3 vaccinations 2 weeks apart followed by 3 monthly injections is currently investigated for ATP128 combined with anti-PD-1 blockade in the ongoing KISIMA-01 clinical trial (Phase 1b Study to Evaluate ATP128, With or Without BI 754091, in Patients With Stage IV Colorectal Cancer - ClinicalTrials.gov).

In various pre-clinical models, the vaccination schedule had been adapted to the fast growth of the mouse tumors to allow for enough time to mount an immune response. The schedule between 2 vaccinations was reduced to one week, and 4 administrations were performed as outlined in detail below. The anti-PD1 treatment was administrated twice a week as usually performed in many studies, and reduced to once a week for the TC-1 model. Regarding the administration of VSV, this was explored regarding scheduling and administration route. As shown in Figure 1, the i.v. route was found the best one after a KISIMA prime. Hence, one viral administration intravenously was performed for all the subsequent experiments.

For the TC-1 tumor model, prime vaccination was administered on day 7 after tumor implantation when palpable tumors were present. The following boost vaccinations were given on day 14, 28 and 49 after tumor implantation. This is consistent with the vaccination schedule applied for testing the immunogenicity of HPV vaccine in non-tumor bearing mice, as the interval between the prime and subsequent boost vaccinations was maintained (Fig 1 and suppl fig 2a-g) - as evidenced in the figure below. In studies depicted in Fig 2, 3 and 4, we analysed the tumor microenvironment changes in TC-1 tumors after 2 total treatments (Day 0 tumor graft – Day 7 prime – Day 14 boost) since the proportion of E7-specific circulating CD8+ T cells is the highest after the 1st boost.

Regimen scheme for HPV vaccine / TC-1 tumor treatment

In contrast, the MC-38 and the EG.7-OVA tumors have faster growth compared to the TC-1 tumor model. Hence, the interval between the prime and subsequent boosts was reduced to 7 days for both models. Importantly, the same interval was

used for both tumor-bearing mice and immunogenicity studies as clarified in the figures below. The 4th vaccination (or Boost 3) was not applied in tumor-free mice in Fig 1h and Fig 1a-g since these studies were performed to test the immunogenicity of the heterologous KVK vaccination against Adpgk (Fig 1h) and OVA (Fig 1a-g). In tumor-bearing mice, we applied a 3rd boost in order to maintain the pool of antigen-specific CD8+ T cells induced by heterologous KVKK combination and prolong the therapeutic window.

Regimen scheme for Mad24 vaccine / MC-38 tumor treatment and OVA vaccine / EG.7 tumor treatment

The following clarifications have been added to the immunization chapter of the Materials and Methods section:

We have added the following to Materials and Methods; page 25

*Vaccination regimens were based on previously published homologous KISIMA vaccinations studies*⁴.

...

...

...

Unless otherwise noted, vaccine treatment intervals in tumor bearing mice followed the application regimen from immunogenicity studies in non-tumor bearing mice for the respective vaccine combinations.

2.

CD8+ T cell functionality is assessed by ex vivo restimulation of cells. Unfortunately, this afford cells (even functionally exhausted ones) an opportunity for additional synthetic restimulation after which they can produce cytokines ex vivo that would otherwise not be produced in vivo (particularly within a suppressive tumor microenvironment). Steps should be taken to assess actual capabilities of these cells (by staining for markers which do not require ex vivo restimulation, such as granzyme B and often IFN-gamma; or by direct analysis after tissue dissection/cell dissociation from animals that received protein export blockade [golgiplug/golgistop] in vivo a few hours prior to euthanasia).

Though ex vivo ELISPOT/ICS are a standard assay to monitor immune response/polyfunctionality in mice as well as in clinical trials, we appreciate this comment and the concern. We would like to emphasize why ex vivo ICS with 6h restimulation was applied in our studies. We also believe that heterologous KVK combination leading to enhanced functional and polyfunctional T cell pools is supported by corroborating data, as shown in details hereafter.

First, regarding the experimental design. In this study we have demonstrated that TC-1 tumors are strongly infiltrated by CD8+ T cells after 1st boost with VSV-GP-HPV in both homologous VV and heterologous vaccinated mice. This has been shown by transcriptome analysis (Fig 3E), flow cytometry analysis (4C) and immunohistochemistry (4D). Importantly, the tumors are infiltrated **by both HPV-E7 (Fig 2D, E) and VSV-N (Suppl fig 3 D, E) specific CD8+ T cells**. Performing intracellular cytokine staining for granzyme B or IFN- γ directly without ex vivo restimulation would not allow us to distinguish specificity of intra-tumoral CD8+ T cells, which is why we turned to ex vivo restimulation.

While ex vivo re-stimulation can induce exhausted T cells to express IFN γ and TNF α , this effect is only partial and can't restore full T cell functionality compared to non-exhausted T cells (see Sandu et al., Nature Communication, 2020; doi: 10.1038/s41467-020-18256-4).

In addition, analyzing the transcriptional levels of freshly harvested tumor tissue from mice treated with different regimen showed higher levels of Ifnyg, Tnfa and Gzmb in KV treated TC-1 tumors compared to VV treated tumors (Fig 3E, F and additional violin plots below). Another molecule associated with cytotoxicity, prf1 (perforin) was also upregulated after KV vaccination compared to VV vaccination (Fig 1E and additional violin plots). Of note, the violin blots were derived from the same data set already presented as a heatmap in the manuscript (Fig 3); to avoid duplicating the data presentation this blot is not included in the revised manuscript but is rather used here to highlight the upregulation of functionality-associated

gene signatures.

Gene expression levels of markers of T cell functionality using nanostring transcriptome analysis of freshly isolated tumor tissue

Although the suggested in vivo golgiplug treatment experiment sounds indeed quite interesting, we were and still are limited by our animal experimentation license. As a license amendment would come with several months processing time, we opted for an ex vivo treatment of TILs with golgiplug in absence of other restimulation (see graphs below), and found that about 80% of tumor infiltrating CD8 T cells in KV treated mice expressed granzyme B (see below).

Finally, to corroborate the transcriptome analysis discussed above, we also added a new study focusing on the protein levels of T cell functionality markers using Luminex assays. Luminex results did not show differences between KV and VV and small difference vs mock, while we do see higher IFN-gamma and TNF-alpha content in KV vs. Mock group (Figure below). These data underline the importance to assess functionality markers in an antigen-specific manner when comparing KV and VV.

We have modified and added the following to Discussion; page 20

Although homologous VSV-GP-TAA also showed efficacy in some models, tumor control was limited. The combination with KISIMA-TAA prime strongly enhanced the proportion of memory T cells, and tumor relapse was delayed. An additional key finding of our heterologous combination was the increase in polyfunctional CD8+ T cells, including IFN- γ , TNF- α , and CD107a as shown with ex vivo intracellular staining. Importantly, the transcriptome analysis of freshly isolated tumor tissue corroborates the notion that heterologous KV vaccination profoundly enhances T cell activity and functionality.

3.

There appear to be some inconsistencies amongst the data that need further investigation and/or consideration. It is mentioned that reduction in viral-specific T cells occurs (in a non-tumor model when compared to OVA) when either KVV or KVK are used (SFig 2i-k). However, in a tumor model (SFig 3b) KV priming appears to increase virus-specific T cells over VV. Also, in Figure 4c it appears that VV dramatically increases T cell infiltrates over KV in flow data, which is shown to be the opposite in 4d histology.

We appreciate these comments and hope to provide helpful clarification here as well as in the manuscript. We have reported the kinetics of the anti-viral T cell response in tumor-bearing mice for 3 different tumor models in supplementary fig 7 B,C, F, G, J, K. It is true that we consistently observed a strong VSV-N specific CD8+ T cell response in the heterologous KVKK group after the boost with the respective VSV-GP-TAA. This is consistent with the results shown in Suppl fig 3b as this analysis was performed after the boost with VSV-GP-HPV. However, long-term (after the 2nd and 3rd boost with the respective KISIMA-TAA) we observed that the proportion of anti-viral CD8+ T cell reduced for all 3 different tumor models analyzed. This is consistent with the data shown in suppl fig 2i-k since this analysis was performed after 2nd boost. It is important to note that the overall reduction of viral-specific T cells in heterologous combination is also evident in enhanced ratio of anti-tumor and anti-viral CD8+ T cells in heterologous KVKK (+/- checkpoint blockade) compared to homologous VVVV vaccination. We generated three additional analysis graphs for each of the applied vaccination models, displaying the ratio of anti-tumor vs antiviral T-cell responses (see figure below). These graphs have been added to supplemental Figure 7. We have rephrased the respective passages in the manuscript and highlighted the KVK

effect on the ratio rather than on antiviral immune responses per se.

Ratio of anti-tumor vs antiviral T-cell responses. Dotted lines indicate time of immunization.

We have added the following to Results; page 17

Surprisingly, circulating anti-viral CD8⁺ T cells were unaffected by checkpoint inhibition in all three tumor models (Supplementary Fig. 7b,c,g,h,l,m) but the ratio of anti-tumor to anti-viral CD8⁺ T cells in circulation was not greatly enhanced by combining checkpoint blockade antibodies with KVK vaccination (Supplementary fig 7d, i, n).

In response to the discrepancy on display between TILs composition and TC-1 tumor histology in Figure 4c vs 4d, we have to admit that an error occurred in the final figure composition. In the flow cytometry TILs data set, KV and VV sets were mixed up. The KV set depicts the data from the VV group and vice versa. This mistake happened while rebuilding the final submission figure to comply with the required RGB style. The figure had originally been composed as a CMYK file. To substantiate the nature of this mistake, we provide the original previous figure draft (see below) as a file with a date from Nov 23rd (manuscript submission was on Dec 9th) as well as the original Graphpad Prism file from Sept. 9th. These files have been uploaded to the submission system and also provided as a zip file (with original date stamp) to the editor.

Figure 4 c has been corrected accordingly.

Originally drafted Fig 4 before CMYK -> RGB conversion. Please note the correct VV vs KV data panel in 4c

4.

It is not clear (here or in the literature) that reducing anti-viral immunity necessarily results in a net positive effect of tumor reduction. Studies have shown previously that increased viral specific T cells in a tumor microenvironment can positively influence tumor growth. While VSV is oncolytic, studies have also shown that a factor in its anti-tumor ability is its immune recruitment. This should be considered and discussed further. (see: Dai et al, Science Immunology, 2017:

<https://immunology.sciencemag.org/content/2/11/eaal1713.long>,

Newman et al, PNAS 2019. <https://www.pnas.org/content/117/2/1119>

Nelson et al, Cell Reports: <https://www.ncbi.nlm.nih.gov/pmc/articles/PMC6874401/>

We would like to thank the reviewer for the recent references and comments, we have re-formulated the following sections in the discussion. Whereas it is established that “turning a cold tumor hot” supports checkpoint inhibition therapy and that oncolytic virus are promoting these changes in the tumor microenvironment as TLR agonists do, we would like to strengthen the importance of balancing antigen specific T cells and bystander T cells. Indeed, removing the break with anti-PD-1 would require the presence of killer T cells able to specifically recognize the tumor cells for clinical efficacy. Hence, in the heterologous prime-boost setting, antigen specific T cells are significantly amplified while innate immunity and anti-viral T cells are contributing to the

immune-promoting changes in the tumor microenvironment. By limiting the injection of the VSV to one administration, the balance is tilting toward antigen specific T cells as shown above in figure in response to comment #3. In addition to these new graphs included in supplemental Figure 7, we modified the respective section in the Introduction, including incorporation of some of the suggested references. Although the discussion already addresses the potentially beneficial role associated with OV-triggered antiviral immune responses (such as discussion page 20 below), a statement was added in the introduction page (as shown below).

Existing section Discussion page 20

“The detection of OVs by the immune system and the resulting mounting of antiviral effects also harbor therapeutic benefits³⁷. Viral infection of tumor tissue results in a rapid innate response, driving a pro-inflammatory environment with subsequent infiltration of antigen presenting cells¹⁰.”

Addition to the Introduction; page 4

One potential limitation is inherently linked to the strong immune activation that comes with two dominant antiviral forces, an initial innate and a subsequent adaptive response^{13, 14}, although these very same mechanisms may also counter tumor associated immune suppression^{15, 16}. Arming OVs with antigens associated with the tumor can additionally enhance the tumor-specific T cell portion and therefore positively affect the balance of antitumor versus antiviral immune responses¹⁷.

5.

Statistical analyses appear to be missing in some panels of figures (including but not limited to Figs 1a, 4b, S1ab, S2ik, S5, S6, S7 some panels, S9d).

Statistical analysis has been completed. The respective sections with additional analyses have been marked in the manuscript:

Modified figures:

Fig 1a – statistical analysis performed, asterisks added

Fig 2f, g, h – black bars added under asterisks to mark groups compared

Fig 4b – statistics was not performed due to n=2 in two groups. Graph was modified to show individual values only

Suppl Fig 1a, b – statistical analysis performed, asterisks added

Suppl Fig 2e, f, i, k – statistical analysis performed, asterisks added

Suppl Fig 3f-i – black bars added under asterisks to mark groups compared

Suppl Fig 5a-d – statistical analysis performed, asterisks added

Suppl Fig 6a – statistical note added

Suppl Fig 6c – no statistical analysis due to n=1 for untreated ctrl; conversion to individual data points

Suppl Fig 6f – statistical analysis performed, asterisks added

Suppl Fig 7 – statistical analysis performed, asterisks added
Suppl Fig 9d – statistical analysis performed
Suppl Fig 10j,k – black bars added under asterisks to mark groups compared

6.

It is unclear whether data shown are only from one experiment (with the n number of mouse listed) or if they are representative of more than one experiment conducted. Multiple experiments would be important for demonstration of reproducibility.

We added information on the number of experimental repetitions to each figure. In brief: for our in vivo experiments, heterologous KISIMA prime and VSV-GP-TAA vaccination was performed in tumor-bearing mice more than once. In vivo study repetitions would also include additional experimental groups, such as immune checkpoint combinations. These additional groups were not always subject to repetition. A detailed description of studies being performed once or multiple times has been added to each figure legend.

7.

Many of the treatment regimens appear to start before tumors are well established, and a prophylactic rationale for such studies is unclear.

As mentioned before (comment 1), the tumor rapid growth of some models allows for a limited therapeutic window in order to mount a potent adaptive immunity. Although the administration of the first vaccination was initiated early in some models (e.g. D3 for the MC38), the vaccination was always performed post-tumor challenge and can not be strictly considered as prophylactic. Prophylactic vaccinations are usually performed D-21 and D-7 before the tumor challenge. It shall be noted that for TC-1 (Vac1 performed at d7) and B16-Ova (Vac1 performed at d5), the tumors were already palpable.

Minor Concerns:

1. All terms should be clearly defined (at least at first use). For example, it is unclear without further searches what antigen Adpgk is, or what type of cell line TC-1 is.

We have added the following to Introduction; page 5

Using various murine tumor models representing different antigen classes (model antigen Ovalbumin or OVA, neoantigens Adpgk (ADP-dependent glucokinase) and Reps1 (RalBP1-associated Eps domain-containing protein 1), and oncoviral antigen E7), we dissect the induced immune components in several compartments addressing immunogenicity and efficacy.

We have added the following to Results; page 6

The immunogenicity of KVK regimen was further assessed against neoantigens and viral oncoprotein as target antigens. For targeting neoepitopes, KISIMA-Mad24, a KISIMA-derived vaccine bearing the previously described neoantigens Adpgk (ADP-dependent glucokinase) and Reps1 (RalBP1-associated Eps domain-containing protein 1) which are expressed in the murine colorectal carcinoma model MC-38²⁸ and the corresponding VSV-GP-Mad24 were used.

We have added the following to Results; page 8

TC-1 cells are transformed murine lung epithelial cells expressing the HPV-derived oncoproteins E6 and E7²⁹.

2. For some figure panels, the groups being compared are unclear (including but not limited to Figs 2f, 2g, S3f,g)

We have revised the figures accordingly to clearly indicate the groups compared. See detailed listing of modified graphs in comment to point # 5

Reviewer #2 (Remarks to the Author):

This well-written manuscript from Das et al reports on a series of preclinical studies that builds on their prior work with a novel chimeric protein cancer vaccine (KISIMA) combined with systemic administration of an oncolytic virus vaccine that encodes the same antigens, in a prime:boost approach. The KISIMA vaccine incorporates a multi-antigenic domain (Mad) with epitopes for CD8 and CD4 T cells + cell penetrating peptide (CPP) + peptide agonist for TLR2 and TLR4 (TLRag). This is abbreviated as K. The viral construct is built on a vesicular stomatitis virus (VSV) variant encoding tumor antigen, and abbreviated as V. The investigators report on extensive preclinical studies in 3 different tumor models, using model tumor antigens, viral tumor antigens, and mutated cancer neoantigens. The prime:boost strategy with KISIMA prior to oncolytic virus (KV), especially with followed with K boost (KVK) induces strong and durable T cell responses in circulation and infiltration of those cells in the tumors. T cell function status is assessed by flow cytometry and gene expression (Nanostring) and reveals both effector and memory T cells, and suggests less exhausted T cells infiltrating tumors when using KVK than other strategies. The tumor control with KVK vaccines is disappointing (1 of 7 tumors with durable control in some experiments), but addition of PD1 blockade (which is ineffective as monotherapy in the models tested) induces durable tumor control in about half of the mice. They also show that IV administration of V is more effective than IM, which is another novel finding. The manuscript includes extensive data in the main figures as well as in supplemental material. The methods are sound and are well-controlled and comprehensive. The work has strong implications for design of optimal cancer vaccines in a range of cancers. There is strong translational potential and high impact across the field of cancer immunology and immunotherapy. The statistical analyses are appropriate and rigorous.

We thank the referee for the positive notion and supportive comments. We have

addressed the points raised below and in the manuscript.

The only criticism is in Figure 4, where the immunohistology images (4d) show higher CD8 densities with KV than with VV, but the data in 4b and 4c suggest that the CD8 infiltrates are likely greater on average with VV. It would be good to modify so that the images in 4d are more representative of the summary data in 4b and 4c, or to explain why there may be a discrepancy.

As addressed above for reviewer 1, point 3:

In response to the discrepancy on display between TILs composition and TC-1 tumor histology in Figure 4c vs 4d, we have to admit that an error occurred in the final figure composition. In the flow cytometry TILs data set, KV and VV sets were mixed up. The KV set depicts the data from the VV group and vice versa. This mistake happened while rebuilding the final submission figure to comply with the required RGB style. The figure had originally been composed as a CMYK file. To substantiate the nature of this mistake, we provide the original previous figure draft (see below) as a file with a date from Nov 23rd (manuscript submission was on Dec 9th) as well as the original Graphpad Prism file from Sept. 9th. These files have been uploaded to the submission system and also provided as a zip file (with original date stamp) to the editor.

Figure 4 c has been corrected accordingly.

One additional question is whether the investigators have compared intratumoral injection of V instead of the IV approach that they have used. In humans, oncolytic viruses have mostly been administered intratumorally. Being able to induce benefit with IV administration is appealing, but it may be good to comment on this in the manuscript and to highlight the advantages of being able to administer IV (for tumors that are not readily accessible percutaneously).

Although VSV-GP initial studies were performed in part with intra-tumoral injection in oncolytic settings without tumor-antigen carrying VSV-GP (Urbiola et al., 2018; Schreiber et al., 2016; Koske et al., 2019), other routes of administration (i.m, s.c. and i.v as shown in Figure 1) were privileged for the development of the heterologous prime-boost toward clinical trial. Even if the approach supporting intra-tumoral injections of oncolytic virus was developed toward accessible tumors like melanoma and head neck as well as administration to liver metastasis, it also reduces the number of possible targeted indications along with narrowing down the number of patients that could be included in the trial. Indeed, the process of intra-tumoral injections for non-superficial tumors is heavy and can be painful for the patients. Hence, the intravenous route was preferred for the virus, which is not only an oncolytic vector in this setting but also vaccine boost.

We have added the following to Discussion; page 20

Whereas KISIMA has routinely been administrated s.c.⁴, oncolytic VSV-GP has been extensively tested in i.v. and intratumoral applications^{16, 19, 34} and the main

route for prophylactic VSV-GP based vaccines has been i.m. ¹⁸.

Reviewer #3 (Remarks to the Author):

1. Das et al. present that the self-adjuvanting cancer vaccine combined with an oncolytic vaccine induces potent antitumor immunity. The authors also show that heterologous vaccination with KISIMA and VSV-GP-TAA could sensitize non-inflamed tumors to checkpoint blockade therapy. The concept of this experiment and the approach for the procedures are designed well on the whole.

We thank the reviewer for the supportive notion and the helpful comments addressed below.

2. The concept of self-adjuvanting cancer vaccine combined with an oncolytic vaccine does not seem to be novel. Can they emphasize what is novel about this study in Introduction or Discussion?

The self-adjuvanting immunization platform KISIMA is a novel recombinant vaccine combining a cell-penetrating peptide and a TLR2/4 peptide agonist, efficacious for the priming of a multiantigenic T cell response, promoting an integrated immune response and memory immunity along with a first reprogramming of the tumor microenvironment. The first KISIMA derived vaccine (ATP128) is currently investigated in a phase 1b in combination with PD-1 blockade in mCRC (Phase 1b Study to Evaluate ATP128, With or Without BI 754091, in Patients With Stage IV Colorectal Cancer - Full Text View - ClinicalTrials.gov). The replacement of the KISIMA-vaccine boost with VSV-GP-TAA boost intravenously injected constitute a novel and unique 3-party combination that shall be entering in clinical trial in the coming months. Oncolytic viruses have been investigated as cancer vaccine platforms in a homologous setting for several years. In addition, heterologous combinations of these oncolytic vaccines with other vaccine partners have been described as well, as we mention in our manuscript introduction and discussion. However, our study pioneers combining an oncolytic vaccine with a non-viral vaccine that is self-adjuvanting. Also, to our knowledge, we provide the most comprehensive in-depth analysis of the mechanisms underlying the therapeutic benefits of the heterologous combination compared to the respective homologous regimens. We built our data collection on multiple antigen and tumor models, employed a broad range of complementing methods to dissect the tumor microenvironment and looked into T cell specificities, quantities, exhaustion status, and functionality. In addition, i.v. application of a vaccine partner is still rather unconventional and we show in our study that the i.v. application of oncolytic vaccines is more beneficial immunologically to the more classical i.m. immunization route.

We have added the following to Introduction; page 5

In addition to a pro-therapeutic TME repolarization, we hypothesize that the heterologous combination of KISIMA-TAA and VSV-GP-TAA not only enhances

the quantity of antigen-specific T cell responses but also affects the quality of the tumor-specific T cell pool on several levels.

3. The order between KISIMA-TAA and VSV-GP-TAA as a priming is important?

We appreciate this important question, which we may not have clearly enough addressed in the initial manuscript version. Indeed, we have observed that the order of KISIMA-TAA and VSV-GP-TAA vaccination is critical. Priming with KISIMA-TAA followed by VSV-GP-TAA boost results in the highest proportion of tumor-antigen specific CD8+ T cell response. In Fig 1a, we demonstrate that KISIMA-OVA prime followed by VSV-GP-OVA boost results in 10-15% OVA-specific CD8+ T cells in periphery in contrast to under 5% circulating OVA-specific CD8+ T cells in mice receiving VSV-GP-OVA prime followed by KISIMA-OVA boost. Importantly, we could further boost the proportion of circulating OVA-specific CD8+ T cells with another dose of KISIMA-OVA (KVK) and not VSV-GP-OVA (KVV). This effect was reproduced when we compared the KVV and VKV vaccine regimen in a tumor-free model with a two-week interval between prime and subsequent boost vaccination as shown in suppl fig 2i. Furthermore, this was reproduced in another study where we immunized non-tumor bearing mice against the Adpgk neoepitope which is expressed in MC-38 tumor cells. Fig 1h summarizes the importance of priming with KISIMA-TAA followed by VSV-GP-TAA boost. We observed 15-fold more Adpgk-specific CD8+ T cells in peripheral blood of KV vaccinated mice compared to VK treated mice 7 days after boost 1. The Adpgk-specific T cells further expanded after a second KISIMA-TAA boost only in mice which had previously received heterologous KV but not heterologous VK. Of note, alternative heterologous vaccine approaches with VSV variants from previous studies also support the application of VSV as a boost, rather than prime. Several of these studies have been referenced and discussed in our manuscript (Pol et al., 2019; Bridle et al., 2010; Atherton et al., 2018). We modified the respective section in the manuscript.

We have added the following to Discussion; page 19

The KV combination generated the strongest CD8⁺ T cell responses when VSV-GP-TAA was applied as a boost compared to a VSV-GP prime. This is consistent with the boosting potential of VSV shown in previous studies using various priming partners^{23, 26, 33}.

4. The authors showed that a single intravenous (i.v.) administration of VSV-GP-OVA induced the highest frequency of antigen-specific CD8+T cell response compared to intraperitoneal (i.p.), subcutaneous (s.c.) or intramuscular (i.m.) injection. However, the i.v. administration only be effective because of its high blood levels.

The virus dose applied was the same in each of the different routes investigated.

However, the referee is correct in stating that after i.v. injection the amount of virus in the blood is higher than with the other routes, naturally. In this regard, it should be considered that within a few minutes, free virus becomes undetectable within the blood (Willmon et al., 2009). In addition, VSV and its variants have been shown to induce strong immune responses via the splenic route (Pol et al, 2014), which supports the findings of enhanced T-cell frequencies after i.v. injection. Finally, the intravenous route of administration explored in mice as shown in Figure 1 was further validated in Non-Human Primate studies and is the intended route of administration in the planned clinical trial.

We have added these considerations to the discussion of the manuscript.

We have added the following to Discussion; page 20

Whereas KISIMA has routinely been administrated s.c.⁴, oncolytic VSV-GP has been extensively tested in i.v. and intratumoral applications^{16, 19, 34} and the main route for prophylactic VSV-GP based vaccines has been i.m.¹⁸. In this study systemic VSV-GP-TAA application was shown to significantly enhance the CD8⁺ T cell responses, without inducing any signs of toxicity. This is in line with previous studies that showed strong adaptive immune responses to VSV variants associated with strong splenic uptake. On the other hand, i.v. application of VSV results in rapid clearance of non-cell bound free virus from the circulation within a few minutes³⁵.

REVIEWERS' COMMENTS

Reviewer #1 (Remarks to the Author):

My major and minor concerns have been adequately addressed by the authors in the revised version of the manuscript. The detailed responses are appreciated.

Reviewer #2 (Remarks to the Author):

The authors of "A modular self-adjuvanting cancer vaccine combined with an oncolytic vaccine induces potent antitumor immunity" have replied thoughtfully to all reviewer comments, have addressed them in the manuscript, and have improved the manuscript as a result. The work offers substantial innovation and provides new data that will help in design of effective vaccines as well as combinations of vaccines with checkpoint blockade therapy.

Reviewer #3 (Remarks to the Author):

The authors have responded to my comments.